



# Carbon-isotope chemostratigraphy, geochemistry, and biostratigraphy of the Paleocene-Eocene Thermal Maximum, deep-water Wilcox Group, Gulf of Mexico (U.S.A.)

Glenn R. Sharman[1], Eugene Szymanski[1,2], Rebecca A. Hackworth[3], Alicia C.M. Kahn[3], Lawrence A. Febo[3], Jordan Oefinger[1], Gunnar M. Gregory[1]

[1]Department of Geosciences, University of Arkansas, Fayetteville, AR 72701, United States of America
[2]Utah Geological Survey, 1594 W. North Temple, Suite 3110, Salt Lake City, UT 84116, United States of America
[3]Chevron Technology Center, 1500 Louisiana St, Houston, TX 77002, United States of America

*Correspondence to*: Glenn R. Sharman (gsharman@uark.edu)

**Abstract**. The Paleocene-Eocene Thermal Maximum (PETM) represents the most pronounced hyperthermal of the Cenozoic era and is hypothesized to have resulted in an intensification of the paleohydrologic cycle, including enhanced seasonality and increased sediment discharge to the coastal ocean. Although the PETM has been widely documented, there are few records
from deposits that form the distal, deep-water components of large sediment routing systems. This study presents new constraints on the stratigraphic placement of the PETM in the deep-water Gulf of Mexico basin through analysis of geochemical, carbon-isotopic, and biostratigraphic data within a ~124 m cored interval of the Wilcox Group. Biostratigraphic and carbon-isotopic data indicate that the PETM extends over ~13.4 m based on acmes in the dinoflagellate *Apectodinium homomorphum* and calcareous nannoplankton *Rhomboaster cuspis* and a ~-2‰ shift in bulk organic δ[13]C values. A decrease
in bioturbation and benthic foraminifera extinction suggest that deoxygenation of Gulf of Mexico bottom waters was coincident with the onset of the PETM. A ~2 m lag in the depositional record separates the onset of the PETM negative carbon isotope excursion (CIE) and deposition of a 5.7 m thick interval of organic-lean claystone and marlstone that reflects a shut-off of the supply of sand, silt, and terrestrial palynomorphs to the basin. An increase in $CaCO_3$ ~4.5 m above the CIE onset is consistent with other sites that indicate ocean acidification and shoaling of the calcite compensation depth during the early PETM.

We interpret deposits of the PETM in the deep-water Gulf of Mexico to reflect the combined effects of increased erosional denudation and rising sea level that resulted in sequestration of sand and silt near the coastline but that allowed delivery of terrigenous mud to the deep-sea. The similarity of oceanographic changes observed in the Gulf of Mexico and Atlantic Ocean during the PETM supports the inference that these water masses were connected during latest Paleocene-earliest Eocene time. Although deposition of typical Wilcox Group facies resumed during and after the PETM recovery, an increased influx of
terrestrial detritus (i.e., pollen, spores, organic debris) relative to marine dinoflagellates is suggestive of long-lasting effects of the PETM. This study illustrates the profound and prolonged effects of climatic warming on even the most distal reaches of large (≥1x10[6] km[2]) sediment routing systems.

## 1 Introduction

The ~56 Ma Paleocene-Eocene Thermal Maximum (PETM) represents the most pronounced episode of global warming of the
Cenozoic Era, with surface temperatures increasing 5-9°C in likely just a few k.y. (Foster et al., 2018; Turner, 2018). Although proxy data suggest a complex and nonuniform climatic response globally, numerous studies have shown that the PETM was associated with profound changes to Earth's hydrologic and sediment routing systems, including enhanced precipitation and seasonality and an associated increase in the caliber and quantity of terrestrial sediment discharged to the coastal ocean (Carmichael et al., 2017 and references within). The PETM is also associated with changes in the global ocean that include



rising sea level (Sluijs et al., 2008b), decreasing seafloor oxygen (particularly on continental shelves and slopes; Carmichael et al., 2017), acidificiation (Penman et al., 2014; Gutjahr et al., 2017), and major ecological shifts (Aubry, 1998; Sluijs et al., 2008a; Carmichael et al. 2017). The PETM likely represents the closest deep-time analog to ongoing and future anthropogenic climate changes (Zeebe and Zachos, 2016; Foster et al., 2018), prompting extensive study since its discovery (Kennett and Stott, 1991).

The global extent, rapid onset, and magnitude of the PETM make this hyperthermal a particularly useful natural case study to examine how changing climatic boundary conditions cause measurable changes in sedimentary parameters through time (i.e., environmental signals; Tofelde et al., 2021) within sediment routing systems. Past research has demonstrated multiple PETM-related signals, including increased rates of siliciclastic sediment production, transport, and deposition (Giusberti et al., 2007; John et al., 2008; Foreman et al., 2012; Pujalte et al., 2015, 2016); changing sediment mineralogy, including an increase in

kaolinite content (Schmitz and Pujalte, 2003; John et al., 2012); and an increase in chemical weathering as inferred from bulk sediment and isotopic weathering proxies (Ravizza et al., 2001; Wieczorek et al., 2013).

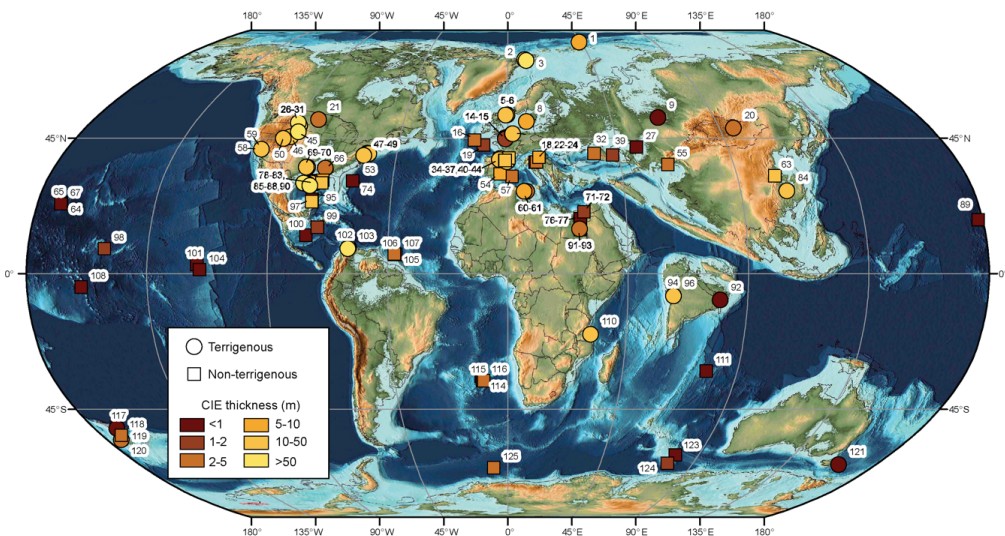

Figure 1. Compilation of PETM localities by dominant sediment source type (terrigenous vs non-terrigenous) and thickness of the PETM CIE. Numbers correspond to site ID (Table S1). Symbols are plotted in order of CIE thickness, such that thicker PETM sections plot on top of thinner ones. CIE thicknesses for Gulf of Mexico sections are based on biostratigraphic data (Cunningham et al., 2022). The global paleogeographic reconstruction is after Scotese (2016).

Although numerous PETM localities have been identified in terrestrial and marine settings, there is a notable lack of documented localities within the distal, deep-water portions of large sediment routing systems (Fig. 1; McInerney and Wing,

2001; Carmichael et al., 2017). Ocean drilling programs have identified multiple non-terrigenous, open-ocean PETM sections with carbon isotope excursion (CIE) thicknesses that range from ~0.1-2.5 m (Fig. 1) but have not yet targeted the PETM within the thick, often sand-rich deposits associated with deep-sea fans (Fig. 1; Table S1). Past research has suggested that large sediment routing systems are likely to buffer pulses of sediment flux if forcing periodicity is less than the timescales over which the upstream river system adjusts to change in boundary conditions (Paola, 1992; Castelltort and van der Dressche, 2003;

Jerolmack and Paola, 2010). For example, Armitage et al. (2011) conducted a numerical experiment that predicted Milankovitch-period cyclicity (20-400 k.y.) to be dampened in most of the world's largest rivers, which have equilibrium time



scales on the order of $10^4$-$10^6$ yr (Castelltort and van der Dressche, 2003). However, several field studies have demonstrated transmission of climate-driven signals within large deep-sea fans, including changing sediment provenance and sediment flux (Goodbred, 2003; Hessler et al., 2018; Mason et al., 2019). Tofelde et al. (2021) suggested that this apparent contradiction may

be related to differences in what is meant by a 'signal' and the fact that a measurable change in a given signal may occur well before the upstream landscape equilibrates. Regardless, identification of the PETM within a large deep-sea fan succession would likely provide a valuable down-system, integrated perspective on the sediment routing system response to abrupt climate-related changes.

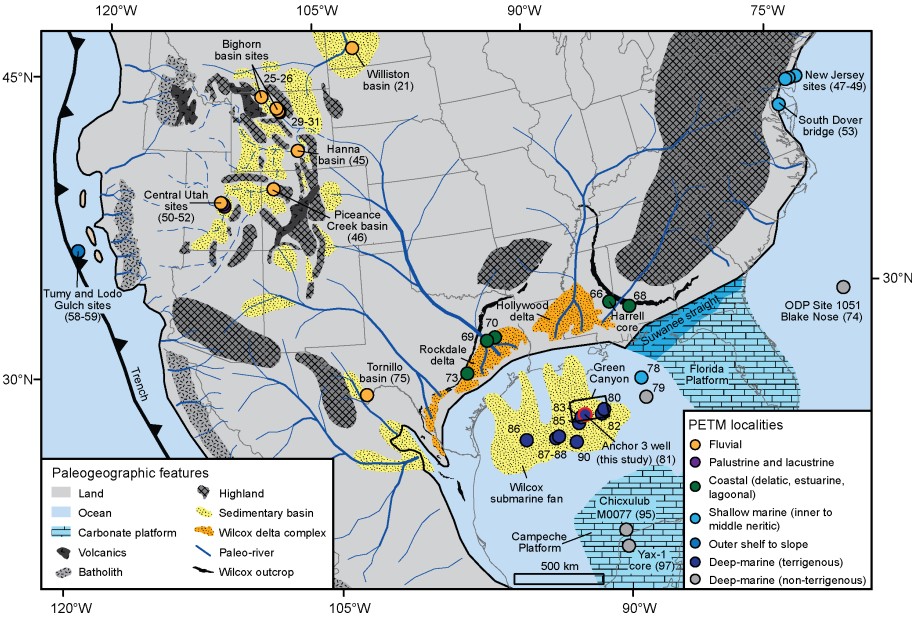

Figure 2. Generalized paleogeography of central North America during Paleocene-Eocene time (modified from Zarra, 2007; Galloway et al., 2011; Sharman et al., 2017; Cunningham et al., 2022). Numbers correspond to PETM localities (Table S1).

This study documents the character of the ~200 k.y. PETM (Westerhold et al., 2018) within deep-sea fan deposits of the Gulf of Mexico (GOM) to clarify the extent to which climate-derived signals are transmitted and/or buffered from source-to-sink and elucidate the effects of the PETM on the Gulf of Mexico and upstream terrestrial environments. We present analysis of ~124 m of drill core from the Anchor 3 well that is located within the Green Canyon protraction area of the GOM basin (Fig. 2; Zarra et al., 2019). The study area is positioned on the distal fringe of a Paleogene continental-scale sediment routing system

that drained ~1-3x$10^6$ km$^2$ of central North America, including well-studied PETM localities within Laramide continental foreland basins (e.g., the Bighorn and Piceance Creek basins) (Fig. 2; Wing et al., 2003; Galloway et al., 2011; Foreman et al., 2012; Sharman et al., 2017; Zhang et al., 2018). Cunningham et al. (2022) recently reviewed publicly available biostratigraphic and geochemical data from GOM wells, primarily from ditch-cuttings samples, and inferred the PETM to range in thickness from 8-222 m. This study aims to determine the lithologic, bio-geochemical, and oceanographic effects of the PETM within

the GOM and Wilcox deep-sea fan system by (1) constraining the stratigraphic position of the PETM through integration of bulk organic carbon-isotopic and biostratigraphic data, (2) evaluating changes in sediment lithology and major element



geochemistry with respect to the timing of the PETM onset, and (3) briefly outlining several geographically and temporally provincial biostratigraphic events that assist in defining the CIE in time. We illustrate how the combined influences of increasing terrigenous sediment supply and rising sea level shape the lithologic and geochemical manifestation of the PETM within the Wilcox deep-sea fan system.

## 2 Geologic background

The Paleocene-Eocene Wilcox Group comprises an expansive set of fluvio-deltaic, shelf, slope, and deep-water depositional sequences within the central and western GOM basin (e.g., Fisher and McGowen, 1967; Edwards, 1981; Galloway et al., 1991a; Xue and Galloway, 1993, 1995; Xue, 1997; Galloway et al., 2000; Zarra, 2007). Derived primarily from the Central and North American Cordillera, these strata contain a 10 m.y. record of Laramide orogenic onset and evolution, continental-scale drainage network reorganization, and deposition of derivative clastic sediment within the GOM basin (e.g., Winker, 1982, 1984; Galloway, 1987, 2008; Galloway et al., 2011). Wilcox Group sediments were delivered to GOM fluvial-deltaic outfall points by drainage systems that operated on multiple scales and reorganized through time (Galloway et al., 2000, 2011 and references therein). With drainage basin scales likely $>1\times10^6$ km$^2$, Wilcox fluvial systems drained a sizeable portion of the North American midcontinent (Sweet and Blum, 2011) and contributed sediment to an immense depositional extent of deep-water turbidites ~100,000 km$^2$ in total area (Meyer et al., 2007). Age equivalent strata extend far into the hinterland where PETM sections are recognized in several Laramide basins (Dickinson et al., 1988; Wing et al., 2003; Bowen and Bowen, 2008; Lawton, 2008; Foreman et al., 2012; Galloway et al., 2011; Kraus et al., 2015; Dechesne et al., 2020). The deep-water Wilcox Group thus represents the terminus of a continental scale sediment routing system with a total fluvial catchment area that was likely comparable to that of the modern-day Mississippi River (Galloway et al., 2011; Blum et al., 2017; Sharman et al., 2017).

Development of the proto-GOM basin began in Triassic-Early Jurassic time with crustal extension and terrigenous sedimentation across a wide zone of continental crust extension before the modern-day Yucatan Peninsula rifted from the rest of Laurentia during a relatively brief period (ca. 166–142 Ma) during the breakup of Pangea (Pindell, 1985; Salvador, 1987, 1991; Buffler and Thomas, 1994; Bird, 2005; Stern and Dickinson, 2010; Minguez et al., 2020; Pindell et al., 2021). From that time onward, the basin captured a nearly continuous depositional record of local and supra-regional tectono-sedimentary events and changing continental landscapes (Galloway, 2008 and references therein). The cessation of oceanic spreading, initiation of rift flank thermal subsidence, and relative dearth of siliciclastic influx during Early Cretaceous time allowed a broad coastal plain and widespread marine carbonate platforms to develop along portions of the North American rift flank (Galloway, 2008). Through most of Cretaceous time, the GOM basin remained relatively underfilled with its margin dominated by extensive carbonate platform and reef accumulations; thin clastic wedges of terrigenous sediment were delivered to the margin via small fluvial catchments that drained rift basin-proximal highlands (Loucks et al., 2017; Galloway, 2008; Snedden et al., 2016a, 2016b; Snedden et al., 2022).



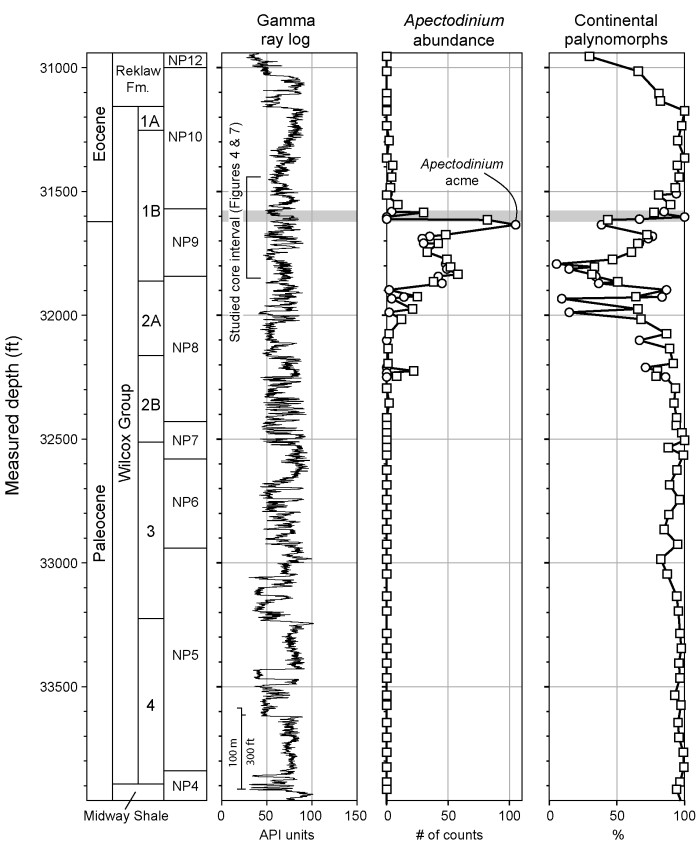

Figure 3. Gamma ray log, total counts of *Apectodinium*, and relative abundance of continental palynomorphs in the Anchor 3 well. Units of the Wilcox Group are from Zarra et al. (2019). Nannoplankton (NP) zones are from this study. An increase in continental palynomorphs (continental spores and pollen grains) relative to marine palynomorphs (dinocysts and acritarchs) may be used to infer changes in terrestrial input into the basin relative to marine productivity. The horizontal gray bar indicates the PETM negative carbon isotope excursion (Fig. 4). Circles indicate core samples and squares indicate cuttings samples. Data are available in Sharman et al. (2022).

The onset and evolution of the Laramide Orogeny during Late Cretaceous and Paleogene time affected the North American

midcontinent tectonically, stratigraphically, and climatically (Dickinson et al., 1988; Feng et al., 1994; Sewall and Sloan, 2006; Copeland et al., 2017; Yonkee and Weil, 2015; Bush et al., 2016) by creating significant hinterland relief and initiating continental-scale drainage reorganization as catchments across the United States and northern Mexico were redirected into the GOM basin (Blum and Pecha, 2014; Lawton et al., 2015; Sharman et al., 2017). Wide basin shelves typically exert a strong control on deep-water sediment budgets by leaving them susceptible to climate-driven sea-level fluctuations (e.g., Sweet and

Blum, 2016), but unprecedented high sediment flux from Laramide hinterland relief generation and catchment enlargement during Paleogene greenhouse conditions overwhelmed typical eustatic and margin architectural controls (Carvajal and Steel, 2006; Carvajal et al., 2009) to carve several large submarine canyon systems (Hoyt, 1959; Galloway et al., 1991b; Galloway, 2007; McDonnell et al., 2008; Brown and Loucks, 2009; Sweet and Blum, 2011; Cornish, 2011; 2019) that received and



channeled a continuous stream of sand-rich siliciclastic sediment from alluvial-deltaic systems to deep-sea fans more than 500

km basinward (Galloway et al., 2000, 2011; Meyer et al., 2007; Zarra, 2007; Blum and Pecha, 2014).

As such, researchers used these observed differences to develop a set of stratigraphic nomenclatures tied to lithostratigraphic packages, seismic sequences, and sediment flux rates (e.g., Lower/Middle/Upper; Galloway et al., 2000) that are useful for source-to-sink system analysis (Snedden et al., 2018; Zhang et al., 2018). Chronostratigraphic frameworks were first developed with inclusion of data from deep-water environs (Zarra et al., 2003; Zarra, 2007) and subsequent refinements to biostratigraphic

zonation and correlation to absolute age constraints from numerous onshore and offshore well data has led to a generally accepted age range for Wilcox Group deposition of 61.5-51.1 Ma, spanning nannofossil biozones NP5 to NP12 with the PETM located near the NP9-NP10 boundary (Zarra et al., 2019).

The Anchor 3 well was drilled in 1443 m of water and reached a total measured depth of 10370 m within the Midway Shale that underlies the ~830 m thick Wilcox Group (Fig. 3). This study focuses on the description of 124 m of drill core that spans

the Paleocene-Eocene boundary, as originally described by Zarra et al. (2019) (Fig. 3). Bedding dips in the interval of core under study are generally ≲10°. Deposition of Wilcox Group turbidites at the Anchor 3 well site during late Paleocene to early Eocene time is estimated to have occurred within the lower portion of upper bathyal to lower bathyal zone (500 to 2000 m) of the continental margin based on benthic foraminiferal assemblages and other considerations (see Sect. 5.3).

### 3 Methods

This study integrates biostratigraphic analysis of both core and ditch-cuttings samples across the entire Wilcox Group (Fig. 3) with focused sampling of core from 31444 to 31852 ft for lithologic, C-isotopic, and major elemental geochemical characterization (Figs. 4 and 5). All sample depths are reported as measured depth in ft, with cutting samples demarcated as a depth range (top to bottom of the cuttings interval). In addition, we assembled an analogue database of geologic attributes from global PETM localities (Table S1) to allow comparison with our results. Sample collection and analytical methods are outlined

below.

### 3.1 Biostratigraphic processing and analyses

Cuttings samples were collected during well-site operations at an average interval of 10 to 30 ft (~3 to ~9 m) for initial biostratigraphic interpretation, and post-well work on the core involved the selection of twenty core plugs. In each case, shale and silt rich intervals were targeted for analysis, expecting optimum fossil recovery. Cuttings and core plugs were processed

and analyzed for foraminifera, nannofossils, and palynology. Biostratigraphic events and zones were selected for sampling based on the presence of both marine fossils and terrestrially derived palynomorphs using range and event stratigraphy and quantitative assemblage patterns of various taxonomic groups. Microfossil data were incorporated into the overall biostratigraphic scheme and are available in Sharman et al. (2022).

Smear slides were prepared for calcareous nannofossil analysis by Ellington and Associates in accordance with standard smear

slide settling techniques (Watkins and Bergen, 2003) and examined at 30 ft (~9 m) resolution through the interval of interest. Core plugs were selected in collaboration with palynological and micro- and nannofossil criteria and analyzed in parallel with the cuttings. Nannofossil analysis was performed using an Olympus BX41 light microscope at 63x-100x magnification, using 10x oculars, 1x optovar, and optical immersion oil. Counts were made from 3 full transects of each slide.



Foraminifera were prepared from twenty core samples using routine preparation techniques for mildly indurated sediments. Approximately 100 g of sediment were soaked in water for 24 hours to soften and disaggregate the material. The samples were then wet sieved over a 63 μm mesh and dried in an oven at 60°C. Residues were then separated into three size fractions (63-180 μm, 180-250 μm, and >250 μm) to facilitate examination. Because diversity and abundances of foraminifera were low, we made absolute counts of the entire sample. Radiolarian abundance counts are relative estimates.

Cuttings and core material were processed for palynological analysis using the standard palynological preparation technique applied for dinocysts and pollen and spore analyses. This procedure involved the removal of all mineral material using hydrochloric acid followed by hydrofluoric acid. The resultant residues were sieved with 20 μm and 10 μm meshes and mounted on three slides per sample, consisting of a dual coverslip for the 10-20 μm and >20 μm size fractions, a >10 μm composite, and an unsieved kerogen slide. Wherever possible, the >10 μm slide was examined for a total combined count of 200 pollen, spores, and dinocysts under 400x magnification. The individual size fractions (10-20 μm and >20 μm) were then examined for rare (<1% of total abundance) palynomorphs that were subsequently included in the overall count and provided further constraint on age.

The ratio of continental to marine palynomorphs is calculated as $C/M = nC/(nC+nM)$, where $nC$ is the number of counts of continental palynomorphs (pollen+spores+freshwater algae) and $nM$ is the number of counts of marine palynomorphs (dinocysts and acritarchs). The C/M ratio plotted alongside the relative abundances of palynomorph species is applied as a proxy to understanding terrestrial influence and sea level fluctuations (e.g., Versteegh, 1994, Tyson, 1995). Kerogen slides were scanned and examined to further assess the type and quality of organic material in the palynology preps.

### 3.2 Lithologic and geochemical characterization

A total of 252 core plug samples were collected over an interval of 124.3 m that spans the Paleocene-Eocene boundary, based on biostratigraphic constraints (Fig. 3). Sample locations were selected using core photographs, with fine-grained intervals sampled more densely (~0.3 m per sample) than those dominated by siltstone and sandstone (~0.6 to 1.0 m per sample). A qualitative description of each sample was created using a stereo microscope and grain-size card to assess grain size, bedding features, and other notable features. Although physical access to the core was not possible due to Covid-19-related travel restrictions, sample descriptions were integrated with high-resolution core photographs (~0.33 mm pixel dimensions) to define lithofacies using the approach of Lowe and Gosh (2004). Core photographs were visually characterized for the degree of bioturbation using the semi-quantitative approach of Droser and Bottjer (1986).

Each core plug sample was screened for major and trace element concentration using an Olympus Vanta M Series portable X-ray fluorescence (pXRF) device. A glass and soil standard (Montana SRM 2711a) were analyzed at the beginning and end of each session to ensure that the instrument was operating within expected parameters. Every 10th core plug was analyzed in triplicate and 26 samples were reanalyzed in a different session to assess variability in results across subsequent analytical sessions. Of the 252 samples, 60 sandstone samples and 60 shale samples that exhibited lithologic and geochemical uniformity were selected for further geochemical characterization via ore-grade laboratory analysis of major element oxide abundance (XRF) and major and trace element concentration (ICE-AES and ICP-MS) using SGS Minerals. Finally, 25 samples were analyzed for X-ray diffraction (XRD) for a qualitative assessment of mineralogy using an Olympus Terra Portable XRD/XRF system (Fig. S1). Lithologic descriptions and geochemical data are provided in Sharman et al. (2022).





The chemical index of alteration (CIA; Nesbit and Young, 1982) was calculated following McLennan (1993) based on molar proportions of major element oxides of Al, Ca, Na, and K:

$$CIA = (\frac{Al_2O_3}{Al_2O_3 + CaO* + Na_2O + K_2O}) \text{ x } 100 \tag{1}$$

Because CaO content may be influenced by Ca in carbonates and phosphates, we applied a correction to estimate CaO contributions from silicate minerals alone, CaO*, following the approach of McLennan (1993). First, CaO contributions were
corrected by assuming that $P_2O_5$ is from apatite, which contributes CaO at a molar ratio of 10:3 relative to $P_2O_5$. The value of $P_2O_5$-corrected moles of CaO was used if less than the number of moles of $Na_2O$; otherwise, the value of $Na_2O$ was adopted as CaO* (McLennan, 1993). Finally, CIA values were discarded from samples that displayed evidence for carbonate cementation, including anomalously high CaO, visual observations of cementation, and XRD data.

**3.3 Carbon-isotope chemostratigraphy**

A total of 157 fine-grained (shale) samples were selected out of the total 252 core plugs for isotopic analysis of bulk organic carbon. A handheld Dremel with a diamond bit was used to sample approximately 0.5-1.0 g of material, with the outermost material discarded to avoid surface contamination. Samples were prepared using a modification of the method used by Suarez et al. (2013). To obtain $\delta^{13}C_{org}$, samples were first dried. If the Dremel did not completely powder the sample, the sample was crushed into a fine powder with a mortar and pestle. Inorganic carbon in the form of carbonate was removed via decarbonation;
10 mL of 3M HCl was added to ∼0.5 – 1.0 g of sample and reacted for ∼2–4 h or until the reaction went to completion. Samples were then rinsed to neutrality with deionized water, dried, and crushed (Suarez et al., 2013).

Because the core samples are from a petroleum field, migrated hydrocarbon was removed via solvent extraction before isotopic analysis. Following the decarbonation process, samples were prepared for hydrocarbon extraction by transferring the dried and crushed decarbonated sample to MarsXpress vessels where 10 mL of hexane ($C_6H_{14}$) and 10 mL of acetone ($C_3H_{60}$) were added
directly to the sample in the vessel. The vessel plug and cap were then placed on the opening of the vessel and torqued to seal tightly. The vessels were placed within the sleeves of the main vessel turntable, distributed evenly in the compartment, and loaded into the CEM MarsXpress Microwave accelerated reaction digestion oven. The samples were heated using the ramp-to-temperature control style at a power level of 800 ramping up to 150°C over 15 minutes and holding at 150°C for an additional 15 minutes. After the microwave completed the 30-minute total run time plus approximately 5 minutes of cool down time, the
samples were poured from the Xpress vessels into hydrolysis tubes with glass microfiber filters placed at the bottom of the tube and attached to a solid phase extraction manifold. The samples were rinsed with additional acetone to maximize the return of the solid sample and ensure the complete rinse of hexane and associated hydrocarbons from the sample. The vacuum pulled the liquids out of the sample leaving a small puck that was transferred to a 1-dram vial to be prepared for carbon isotopic analysis.




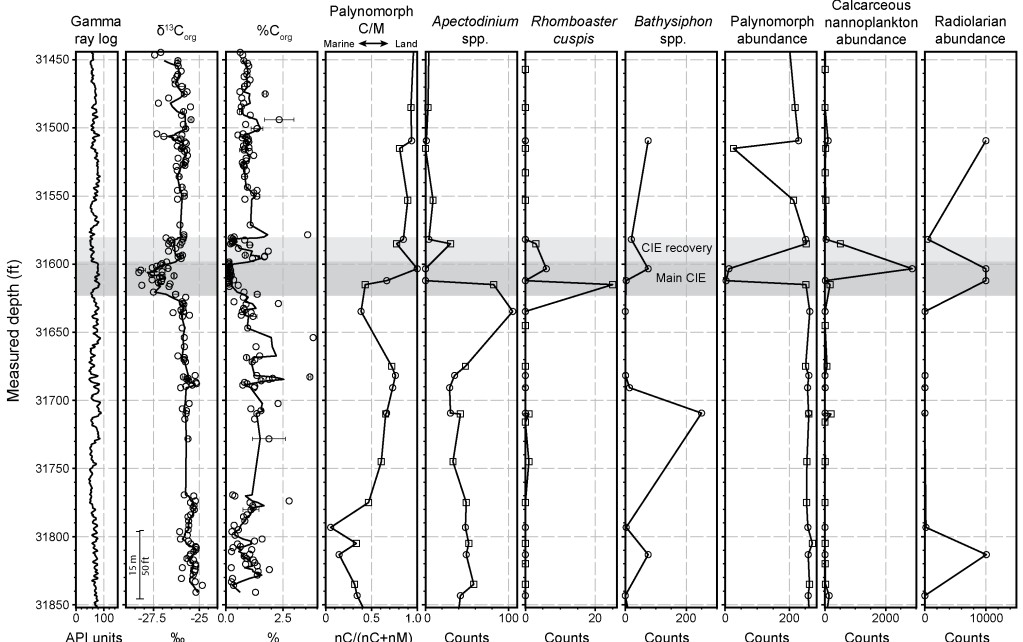

Figure 4. Summary of C-isotopic and biostratigraphic results (data available in Sharman et al., 2022). Solid lines in $\delta^{13}C_{org}$ and %$C_{org}$ plots indicate a 3-point running average. Palynomorph C/M ratio is expressed as $nC/(nC+nM)$, where $nC$ is the number of continental palynomorph specimens counted (continental spores + pollen grains) and $nM$ is the number of marine palynomorph specimens counted (marine dinocysts + acritarchs). Only core samples are shown for *Bathysiphon* spp. and radiolarian counts due to poor preservation in cuttings samples. The main and recovery phases of the CIE are shown by dark and light gray shading, respectively. Core samples are shown as circles and cuttings samples are shown as squares with the mid-point of the cuttings depth range shown.


Following the hydrocarbon extraction process, ~0.5 g of crushed sample was weighed into tin capsules using a Mettler Toledo DeltaRange microbalance with an accuracy of ± 0.002 mg for analysis on a ThermoFinnigan Delta Advantage isotope ratio mass spectrometer (IRMS) via combustion in an elemental analyzer at the University of Arkansas Stable Isotope Laboratory. Values of $\delta^{13}C_{org}$ are recorded in per mil (‰) relative to VPDB (Vienna Pee Dee Belemnite) and are calculated with the use of

laboratory standards. Instrument stability, accuracy, and precision are also monitored via standard analysis. Results from the IRMS were calibrated using the following internal and international standards: Corn Maize (-11.33 ± 0.11‰, actual = -11.32‰), White River Trout (-26.60 ± 0.15‰, actual = -26.63‰), Benzoic Acid (-27.84 ± 0.04‰, actual = -27.64‰), and ANU Sucrose (-10.46 ± 0.05‰, actual = -10.4‰). Total organic carbon values within the decarbonated samples were calculated using Sandy Soil (-25.95 ± 0.49‰, %C = 0.83 ± 0.07) following the methods of Suarez et al. (2013). Carbon-isotope data may

be accessed via Sharman et al. (2022).

### 3.4 Compilation of PETM sections

To place the Anchor 3 core in context with other PETM sections globally, we compiled a list of PETM sections using existing compilations as a starting point (Table S1; McInerney and Wing, 2001; Carmichael et al., 2017). The thickness of the PETM CIE was measured, where available. Each locality was assigned a depositional setting classification (fluvial, lacustrine, coastal,

inner-middle shelf, outer shelf to slope, or deep-marine); the pre-PETM depositional setting was used if a change in depositional setting occurred during the PETM (e.g., Pujalte et al., 2016). PETM sections were further classified as terrigenous if pre-PETM



lithologies are dominated by siliciclastic lithologies (e.g., sandstone, siltstone, clay-rich mudstone) or non-terrigenous if pre-CIE lithologies are dominated by non-siliciclastic, typically biogenic lithologies (e.g., limestone, marl, chalk, pelagic ooze) that reflect starvation of clastic sediment. Lithology change during the PETM was recorded and classified as a grain size

increase or decrease (terrigenous localities) or as an increase or decrease in terrestrial sediment relative to carbonate or biogenic sediment (non-terrigenous localities). Paleogeographic coordinates were reconstructed using the PALEOMAP PaleoAtlas for GPlates (Scotese, 2016).

## 4 Results

### 4.1 Carbon isotope chemostratigraphy

A total of 183 measurements of $\delta^{13}C_{org}$ were collected from 156 depth intervals (Fig. 4). Values of $^{13}C_{org}$ display relatively little variation (-24.8‰ to -26.4‰; average of -25.6‰) in the lower 66 m of the studied interval (samples 249 through 129), with an average total organic carbon (TOC) value of 1.18%. A negative carbon isotope excursion (CIE) occurs between sample 130 (-25.7‰) and sample 128 (-27.5‰) and extends upwards for ~13 m, with the lightest $\delta^{13}C_{org}$ values occurring in the bottom ~7.3 m, herein termed the main CIE (Fig. 5). In the main CIE, the 3-point moving average of $\delta^{13}C_{org}$ is up to ~2‰ lighter than the

pre-CIE average of -25.6‰. TOC values decrease rapidly following the onset of the CIE before reaching consistently low values (average of 0.17%) between samples 121 and 104. An interval with $\delta^{13}C_{org}$ values that are intermediate between the main CIE and pre-CIE values extends upwards for an additional ~6 m (samples 102 to 83) with an average value of -26.2‰, herein termed the CIE recovery. TOC is variable in this interval, with some samples displaying values nearly as low as in the underlying negative CIE (e.g., average of 0.35% from samples 89 to 82) whereas other samples have TOC more typical of pre-

CIE values (average of 1.2% from samples 102-90). The upper 41 m of core (samples 82 through 1) display relatively consistent $\delta^{13}C_{org}$ values with an average of -26.1‰, approximately 0.5‰ lighter than pre-CIE values (Fig. 4). Seven samples in this interval yielded $\delta^{13}C_{org}$ less than -26.5‰, but these samples, and many others from the upper 19 m of core, come from intrabasinal (Figs. 6 and 7). TOC values above the CIE recovery yield an average of 1.0%, similar to pre-CIE values (Fig. 4).

### 4.2 Lithology, lithofacies, and bioturbation

Six lithofacies (LF-1 through LF-6) were defined based on core sample descriptions and visual interpretation of core photographs (Table 1; Fig. 6). LF-2 is the most abundant lithofacies, comprising 46% of the studied interval, and is composed of thinly to moderately bedded mudstone and siltstone, with lesser thin beds of lower very fine sandstone (Figs. 6 and 7; Table 1). LF-2 is distributed below and above the CIE, and within the lowermost ~1 m of the main CIE, including the onset of the excursion between samples 130 and 128. Mudstone within LF-2 has a median TOC of 0.93%. Although carbonaceous (organic)

debris is common in LF-2, it is more abundant above the CIE than below. The CIE recovery is exclusively composed of LF-2.

LF-4, moderately to thickly bedded coarse siltstone and sandstone, is the next most abundant lithofacies (33% of the studied interval) and is found exclusively below or above the CIE except for a 0.4 m interval in the lowermost CIE (Figs. 6 and 7). LF-4 is typically either massive or planar laminated. Below the CIE, LF-4 often overlies LF-2 in crude coarsening-upwards packages as can be observed on the gamma ray log. Above the CIE, LF-4 tends to overlie LF-2 across abrupt contacts while

displaying fining-upwards packages. The upper ~19 m of studied core consists of alternating LF-4 and LF-5 (sand-matrix mudclast conglomerate), the latter of which is found nearly exclusively above the CIE (Fig. 7). The organic content and lithologic character of the mudclasts is very similar to LF-2.



Table 1. Description of lithofacies

| Lithofacies | Subtype | Grain size | % organic carbon median (interquartile range) | Bedding character, sedimentary structures, bioturbation, and other notable features | Stratigraphic position relative to CIE |
|---|---|---|---|---|---|
| Lf-1: Organic-lean claystone and marlstone with subordinate interbedded siltstone and rare sandstone | Lf-1a (non-calcareous) | Dominantly claystone with interbedded fine siltstone and rare lower very fine sandstone. | 0.15% (0.14%-0.18%) N=11 analyses | Finely laminated to locally massive bedding. Dominantly fissile, but locally indurated. Noticeable lack of large (>1 cm) burrows, but some intervals appear to have concentrations of small (mm-scale), horizontal burrows. | Exclusively found within the lower main CIE. |
| | Lf-1b (calcareous) | Calcareous claystone (marlstone) and mudstone. | 0.17% (0.14%-0.19%) N=14 analyses | Finely laminated to massive bedding. Dominantly fissile, but locally indurated. Zones of indistinct bedding appear to be bioturbated. Occasional large (>1 cm) horizontal burrow in upper portion. Distinctively calcareous relative to Lf-1a. | Exclusively found within the upper main CIE. |
| LF-2: Thinly to moderately bedded mudstone and siltstone, with lesser thinly bedded sandstone | NA | Dominantly mudstone, silty mudstone, muddy siltstone, and siltstone. Minor (<5%) lower very fine sandstone. | 0.93% (0.72%-1.28%) N=125 analyses | Thinly (<10 cm) to moderately (10-50 cm) bedded. Dominantly indurated, but locally fissile. Commonly laminated with abundant ripple cross-laminations in siltstone and lower very fine sandstone. Carbonaceous debris becomes common in the upper half of the core. Bioturbation is common. | Found within the lowermost CIE. Common below and above the CIE. Constitutes the entirety of the CIE recovery. |
| LF-3: Thinly to moderately bedded siltstone and sandstone, with lesser mudstone | NA | Coarse siltstone to upper very fine sandstone. Minor (<10%) mudstone. | NA | Thinly (<10 cm) to moderately (10-50 cm) bedded. Indurated. Pervasively current-structured, with abundant ripple cross-laminations. Local soft-sediment deformation. Bioturbation is present but less common than in LF-2. | Found within the lowermost main CIE. Also found locally below and above the CIE. |
| LF-4: Moderately to thickly bedded coarse siltstone and sandstone | NA | Coarse siltstone to lower very fine sandstone. | 0.74% (0.72%-0.82%) N=7 analyses | Moderately (10-50 cm) to thickly (>50 cm) bedded. Indurated. Massive to planar laminated. Generally well-sorted. Locally contains mudstone, becoming gradational with LF-5. Local soft-sediment deformation. Rare bioturbation. | Found within the lowermost main CIE. Common below and above the CIE, including immediately above the CIE recovery. |
| LF-5: Sand-matrix mudclast conglomerate | NA | Mudstone, muddy siltstone, and silty mudstone clasts within a matrix of coarse siltstone to lower very fine sandstone. | 0.95% (0.79%-1.21%) N=22 analyses | Moderately (10-50 cm) to thickly (>50 cm) bedded. Indurated. Massive with crudely stratified with rounded to angular mudclasts. Mudclasts often display laminated and contorted bedding. Rare bioturbation. | Found exclusively above the CIE in association with LF-4. |
| LF-6: Chaotically bedded units, dominantly mudstone | NA | Intermixed silty mudstone and muddy siltstone with lesser coarse siltstone to lower very fine sandstone. | 1.21% (1.09%-1.85%) N=4 analyses | Chaotically bedded units up to 1.4 m in thickness. Abundant mudstone clasts, sometimes folded and micro-faulted, within a matrix of mudstone and some intermixed sand. Also includes isolated, deformed sandstone beds and clasts. | Only found in one occurrence below the CIE and two occurrences above the CIE. |





LF-1 accounts for ~5% of the total interval studied and consists of claystone and marlstone, with subordinate interbedded siltstone and rare very fine sandstone (Figs. 6 and 7). LF-1 is further subdivided into non-calcareous (LF-1a) and calcareous (LF-1b) subtypes. Both LF-1a and LF-1b are exclusively found within the main CIE and constitute the majority (~77%) of this interval. LF-1 is anomalous relative to other lithofacies in its (1) abundance of clay and corresponding lack of silt and sand, (2) low organic carbon content (median of 0.15% to 0.17% for LF-1a and LF-1b,

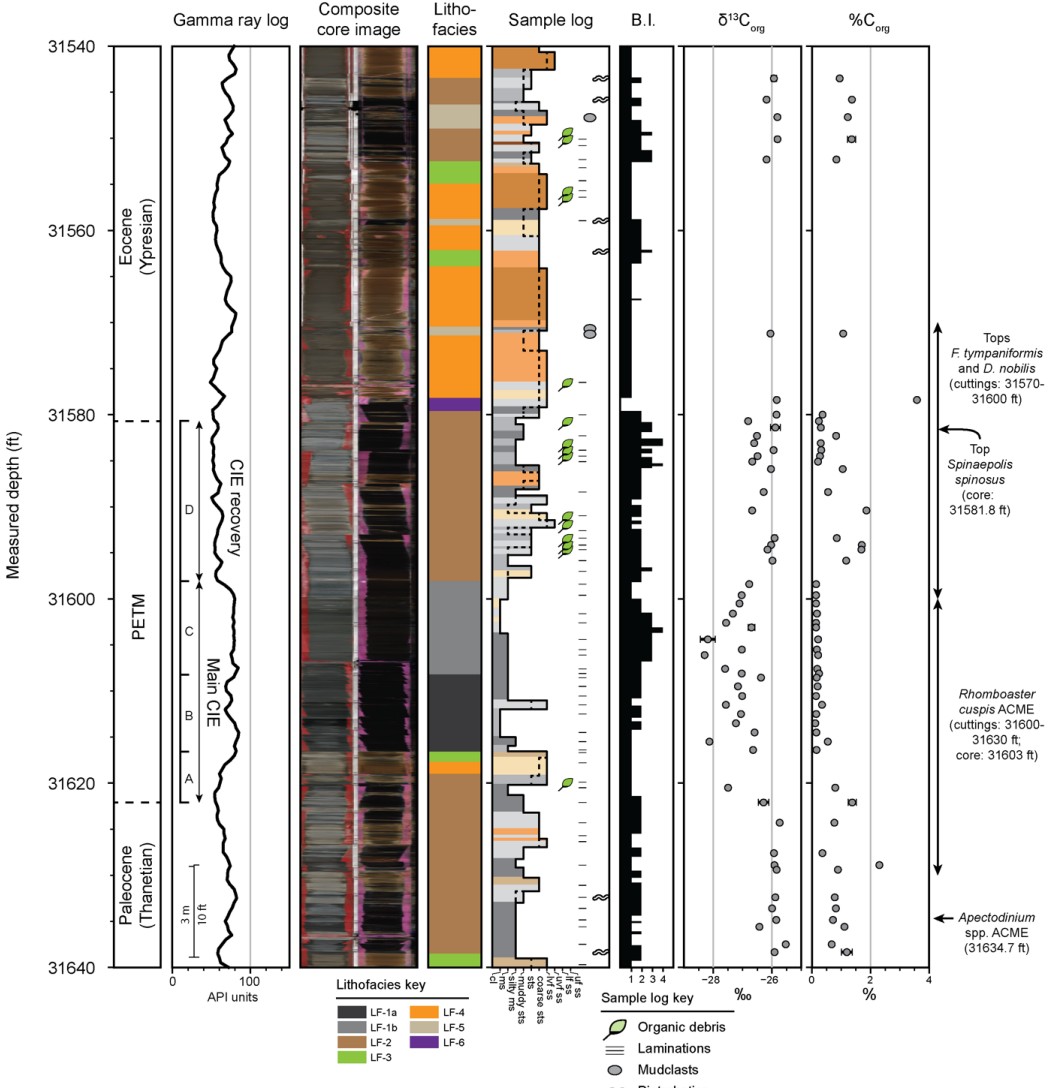

Figure 5. Summary of lithologic and carbon-isotopic results across the Paleocene-Eocene boundary. The dashed and solid black lines in the sample log represent lower and upper grain sizes present, respectively. Units of the main and recovery phases of the CIE are shown on the gamma ray log plot and are discussed in the text. Abbreviations: B.I.-bioturbation index; cl-clay; ms-mudstone; sts-siltstone; ss-sandstone; lvf-lower very fine; uvf-upper very fine; lf-lower fine; uf-upper fine.

respectively), and (3) associated lack of carbonaceous (organic) debris. LF-1b is the only lithofacies in the described interval
to have a significant component of calcite, which is manifested as increased CaO abundance (Fig. 7).





LF-3 comprises thinly to moderately bedded siltstone and sandstone that is pervasively current structured, often with ripple cross-lamination (Fig. 6). LF-3 occurs in isolated intervals typically associated with either LF-2 and/or LF-4 and accounts for ~5% of the studied section. A thin interval of LF-3 (~0.3 m) occurs immediately below LF-1a within the base of the main CIE (Fig. 7).

LF-6, chaotically bedded units, is the least abundant lithofacies and only occurs in three discrete intervals that range in thickness from 0.4 to 1.4 m (Figs. 6 and 7). LF-6 is dominantly composed of deformed mudstone clasts within a mudstone matrix, with lesser amounts of deformed sandstone clasts and sand grains within the mud matrix.

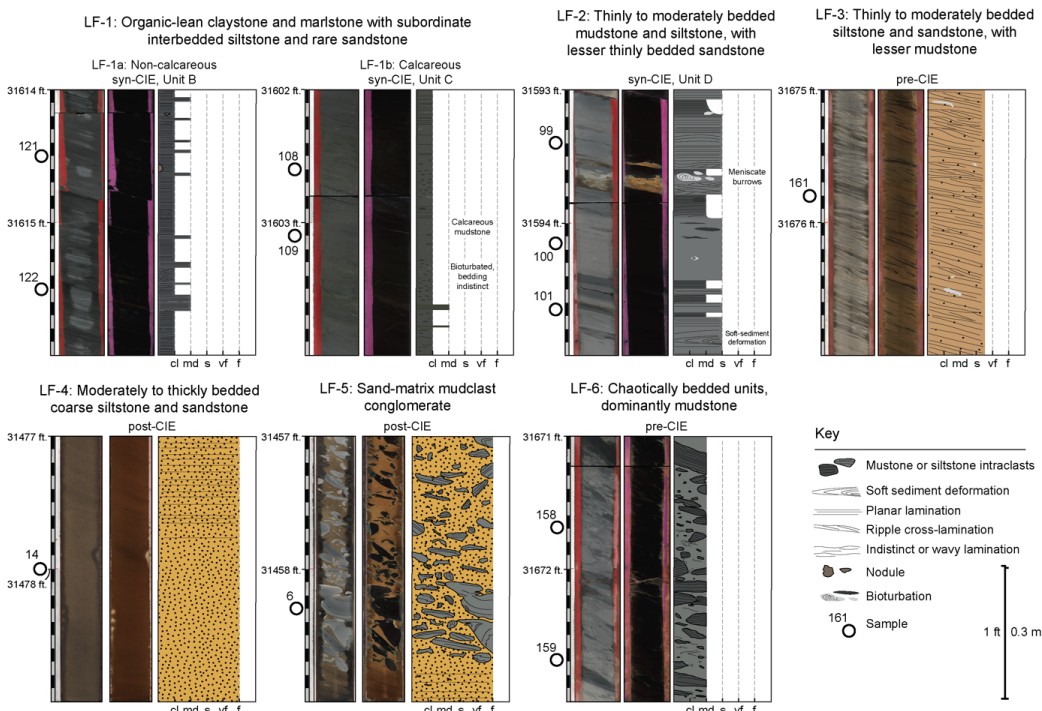

Figure 6. Representative core photograph and drafted section of each lithofacies (Table 1). Core photographs are shown with plain light on the left and ultraviolet light on the right. Numbered white circles correspond to sample number. Location of the photograph with respect to the PETM CIE is identified (see Figs. 4 and 5). White and black scale marks indicate divisions of 0.1 ft (~0.03 m). Abbreviations: cl-clay; md-mud; s-silt; vf-very fine sand; f-fine sand.

**4.3 Bioturbation**

Although the bioturbation index ranges from 1 to 4 (Fig. 8), most of the studied interval was assigned an index of either 1 or 2 (Fig. 7). Bioturbation is more common in LF-2 and LF-3 and is less common in LF-4, -5, and -6. There is a notable lack of bioturbation in the lower ~2.4 m of the main CIE, despite favorable lithologies (LF-2, LF-3, and LF-1a). The bioturbation index then increases upwards within the main CIE to maximum values within LF-1b and remains high in the CIE recovery section (Fig. 7).


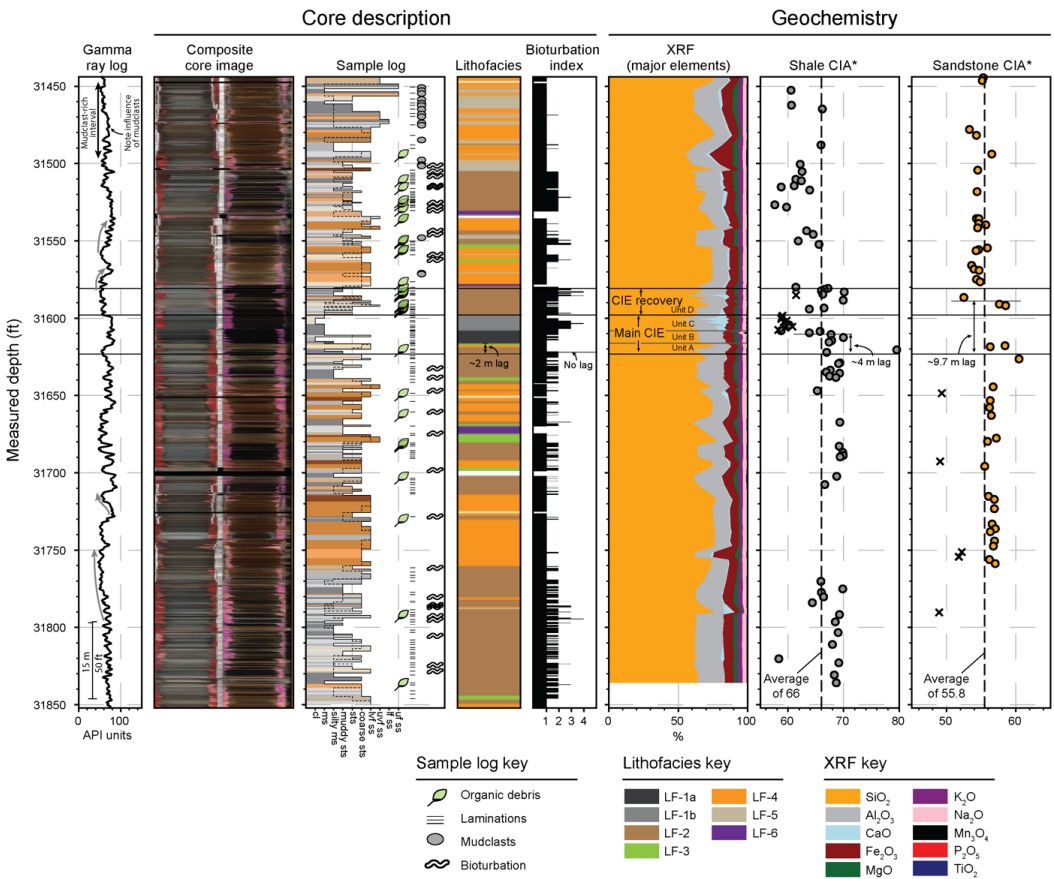

Figure 7. Summary of lithologic and geochemical results. Gray arrows next to the gamma ray log indicate interpreted coarsening- and fining-upwards packages. The dashed and solid black lines in the sample log represent lower and upper grain sizes, respectively. The main and recovery phases of the CIE are shown by horizontal, black lines. Boundaries between Units A-D are shown on the XRF panel. Lags between the onset of the PETM CIE and the associated lithologic or geochemical change are noted. Samples that display evidence for carbonate minerals are shown by black 'x's and are not included in the average CIA* values noted by the vertical, dashed black line. Abbreviations: cl-clay; ms-mudstone; sts-siltstone; ss-sandstone; lvf-lower very fine; uvf-upper very fine; lf-lower fine; uf-upper fine.

## 4.4 Biostratigraphic framework

Biostratigraphic analysis on well cuttings provided the initial chronostratigraphic framework utilizing calcareous nannofossil, foraminiferal, and palynological events to constrain the studied well interval to early Paleocene (Danian) to early Eocene (Ypresian ) time with the cored interval representing late Paleocene (Thanetian) to earliest Eocene (Ypresian) time; more precisely, NP8 to NP10 nannofossil zones (Zarra et al., 2019) (Fig. 3). Biostratigraphic events are represented by highest occurrence (Top), lowest occurrence (Base), and high taxon abundances (common or Acme events). Significant events



documented in the core and the cuttings are listed in Table 2. Note that fossil abundance varied between cutting and core
samples, lithologies, and disciplines.

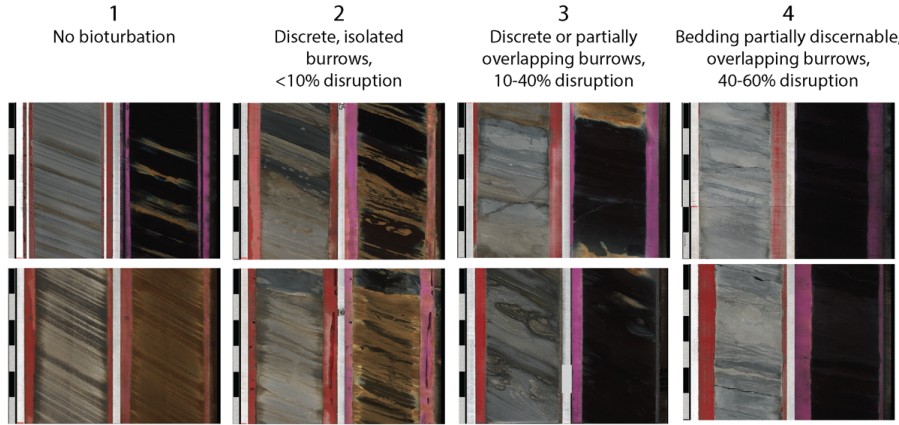

Figure 8. Examples of different bioturbation indices in core photographs, with plain light on the left and ultraviolet light on the right. Bioturbation index values follow Droser and Bottjer (1986). Black and white scale bar shows units of 0.1 ft (~0.03 m).

### 4.4.1 Nannofossils

Calcareous nannofossil assemblages are characterized by relatively low abundances in late Paleocene time, with an increase in
abundance across the Paleocene-Eocene boundary and into the Eocene (Fig. 4). The CIE section is characterized by higher
abundances and more diverse assemblages, including excursion taxa which are discussed below. Despite the low abundance
sections, the major nannofossil markers were in stratigraphic order and integrated with palynological, foraminiferal, and overall
assemblage data to yield a robust biostratigraphic sequence (Fig. 4). *Fasciculithus involutus* and *Fasciculithus tympaniformis*
(31570-31600 ft) are consistently present during the negative CIE with Tops slightly younger than the recovery interval,
indicating lowermost NP10 to uppermost NP9. Zone NP9 includes the CIE and is bracketed by Tops *Discoaster diastypus* and
*Rhomboaster cuspis* (31570-31600 ft), an acme of *Fasciculithus tympaniformis* (31603.35 ft), a single occurrence of *Discoaster
araneus* (31660-31690 ft), and Top *R. cuspis* (31730-31760 ft). Although nannofossils are sparse in abundance in pre-CIE
sections, several events within the core define the basal section of zone NP9 including Top *Discoaster multiradiatus* (common)
and Top *Helicosphaera kleinpelli* and *Helicosphaera riedelii* (32340-32370 ft).

**4.4.2 Foraminifera**

Planktonic foraminifera are rare in the studied core and cutting samples. A cutting sample from 31570-31600 ft contains a
single specimen of *Subbotina velascoensis*, the highest occurrence of which indicates the lower part of zone E1 (Ypresian). A
core sample at 32233.25 ft contains single specimens of *Globanomalina pseudomenardii, Morozovella angulata,* and
*Morozovella conicotruncata*, plus several specimens of *Igorina pusilla* and *Subbotina triloculinoides*. The overlapping ranges
of these taxa indicate zone P4b (Thanetian) as defined by Berggren and Pearson (2005, 2006) and Zarra et al. (2019). Two
arenaceous events are noteworthy: the first downhole occurrences of abundant *Bathysiphon eocenica* at 31690.7 ft and
*Rzehakina epigona* at 31926.4 ft, both of which occur in late Paleocene time at approximately 56.5 Ma (Table 2).





Table 2. Key biostratigraphic events recorded in GOM Anchor 3 well

| Event | Description | Depth (ft) | Sample source | Biozone | Calibrated Age (Ma) using GTS 2012 |
|---|---|---|---|---|---|
| *Morozovella subbotina** Top | ---- | 30940-30970 | cuttings | NP12 | 50.67 |
| *Thomsonipollis magnificus** HRO | Highest regular occurrence | 31000-31030 | cuttings | ---- | ---- |
| *Discoaster multiradiatus* Top | ---- | 31030-31060 | cuttings | NP10 | 53.7 |
| *Discoaster lenticularis** Top | ---- | 31030-31060 | cuttings | ---- | 54.17 |
| *Tribrachiatus contortus* Top | ---- | 31030-31060 | cuttings | ---- | 54.12 |
| *Fasciculithus* spp. Top | ---- | 31500-31530 | cuttings | NP10 | ---- |
| *Bathysiphon* spp. increase | ---- | 31570-31600 | cuttings | ---- | ---- |
| *Fasciculithus involutus** Top | ---- | 31530-31536 | cuttings | ---- | 55.6 |
| *Fasciculithus tympaniformis* Top | ---- | 31570-31600 | cuttings | ---- | 55.5 |
| *Discoaster nobilis** Top | ---- | 31570-31600 | cuttings | NP9 | 55.86 |
| *Spinaepollis spinosus* Top | ---- | 31581.8 | core | ---- | ---- |
| *Rhomboaster bramlettei* Base | First common ocurrence | 31600-31630 | cuttings | NP9 | 55.9 |
| *Apectodinium augustum* Top | ---- | 31600-31630 | cuttings | ---- | ---- |
| Peak *Rhomboaster cuspis* and *Rhombaster* spp. | PETM excursion taxa | 31600-31630 | cuttings | ---- | ---- |
| *Apectodinium homomorphum* ACME Top | Decline in *Apectodinium homorphum* ACME event, decrease in overall *Apectodinium* spp. abundance | 31600-31630 | cuttings | ---- | ---- |
| *Fasciculithus clinatus* Top | | 31603.35 | core | ---- | 55.86 |
| Radiolarian peak | ---- | 31612.12 | core | ---- | ---- |
| Onset of CIE* | ---- | 31623.2 | core | ---- | 56.01 |
| *Apectodinium homomorphum* and *Apectodinium* spp. ACME | Peak *Apectodinium* spp. and *Apectodinium homorphum* ACME | 31634.7 | core | ---- | 55.96 |
| *Apectodinium augustum* Base | Seen with single *Discoaster araneus* | 31660-31690 | cuttings | ---- | ---- |
| *Holkopollenites chemardensis* Top | ---- | 31660-31690 | cuttings | ---- | ---- |
| *Lanagiopollis lihoka* Top | ---- | 31681.7 | core | ---- | ---- |
| *Heliolithus kleinpellii** Top | ---- | 31660-31690 (reworked), 31843.35 | cuttings, core | NP9 | 57.21 |
| *Eocladopyxis peniculata* peak | Peak in dinocysts abundance and diversity w/ *Eocladopyxsis peniculata* and *Spiniferites* complex | 31813.2 | core | ---- | ---- |
| *Discoaster multiradiatus* Base | Lowest common | 31926.4 | core | NP9/NP8 | ---- |
| *Rzehakina epigona* Top | ---- | 31926.4 | core | ---- | ~56.5 |
| *Bathysiphon eocenica* INCR | ---- | 31960.7 | core | ---- | ~56.5 |
| *Insulapollenites rugulatus* Top | ---- | 31960-31990 | cuttings | in NP8 | ---- |
| *Momipites actinus** Top | ---- | 31988.34 | core | in NP8 | 58.1 |
| *Globanomalina psuedomenardii* Top | Very rare, low stratigraphically | 32233.25 | core | ---- | 57.1 |
| *Morozovella angulata** Top | With *M. conicotruncata*, very rare | 32233.25 | core | in NP8 | 58.32 |
| *Igorina pusilla** Top | ---- | 32233.25 | core | ---- | 58.44 |





| | | | | | | |
|---|---|---|---|---|---|---|
| *Heliolithus riedelii* Base | ---- | 32370-32400 | cuttings | ---- | 58.7 |
| *Bomolithus elegans* Top | Low stratigraphically | 32520-32550 | cuttings | NP8 | 58.4 |
| *Fasciculithus janii** Top | ---- | 32820-32850 | cuttings | ---- | 59.3 |
| *Caryapollenites veripites* LDCO | Last downhole common occurrence | 32910-32940 | cuttings | in NP5 | ---- |
| *Fasciculithus ulii* Top | ---- | 32970-33000 | cuttings | ---- | ---- |
| *Fasciculithus magnicordis* Top | ---- | 33240-33270 | cuttings | in NP5 | 60 |
| *Chaismolithus danicus* Top | Rare, low stratigaphically | 33480-33510 | cuttings | in NP6 | ---- |
| *Cruciplacolithus edwardsii* Top | ---- | 33720-33750 | cuttings | ---- | 61.16 |
| *Fasciculithus typaniformis* Base | ---- | 33840-33870 | cuttings | ---- | 61.51 |

*Denotes an event that is used in calculation of average sedimentation rates

LDCO = last downhole common occurrence

HRO = highest regular occurrence

INCR = increase in abundance

### 4.4.3 Palynomorphs

Terrestrial and marine palynomorphs fluctuate throughout the Paleogene and range from poor to excellent fossil recovery and preservation (Fig. 4). Key marker taxa and events were calibrated with calcareous nannofossil and foraminiferal events from GTS2016 (Gradstein et al., 2016) and regional GOM events from previous studies (Wing et al., 2005; Hackworth et al., 2018; Zarra et al., 2019). The occurrence of *Apectodinium homomorphum* ACME event and rare Top *Apectodinium augustum* at 31634.75 ft and 31600-31630 ft, respectively, constrain the PETM interval (Fig. 4). This event is further constrained, both in

cuttings and core, by the following terrestrial pollen extinction events including Tops *Spinaepollis spinosus* (31581.8 ft), *Lanagiopollis cribellatus* (31570-31600 ft), *Holkopollenites chemardensis* (31600-31630 ft), and *Retitricolpites anguloluminosus* (31660-31690 ft).

### 4.4.4 Best fit depth-age interpretation (pre- to post-CIE)

The Tops of biostratigraphically significant microfauna are detailed in the preceding paragraphs, but these Tops do not

necessarily occur at their first and last appearance datums. A best-fit line of correlation was made through as many of the events as possible to determine a preferred age-depth model for the Anchor 3 well. Table 2 lists the biostratigraphic events for the cored interval as well as cuttings from the Wilcox portion of the well. The depths, ages, and event names that were honored in the line of correlation are denoted with an asterix. An important distinction in the results below is usage of the phrase "rock accumulation rates" instead of "linear sedimentation rates". Rock accumulation rates represent average rates over a depth

interval which may include condensed or eroded intervals that we are unable to resolve within the given framework (Carney et al., 1995). In contrast, linear sedimentation rates are instantaneous rates that are typically used in decompacted sections and within the context of shallow, late Pleistocene to Holocene sediment cores where detailed absolute ages are achievable with the precision of radiocarbon and radiogenic nuclides (Sadler, 1981).

Rock accumulation rates in the Anchor 3 Wilcox are generally rapid due to the accumulation of thick, sandy intervals, averaging

at ~0.12 m/k.y. However, the 66 m interval below the CIE from the Top of *H. kleinpellii* (31843.35 ft) to the onset of the CIE



at ~31623 ft shows a lower accumulation rate of ~0.056 m/k.y. Rock accumulation rates increase to ~0.065 m/k.y. from the onset of the CIE through the recovery phase (Table S2). The 12 m interval above the recovery phase, from 31570 ft to 31530 ft (Tops *Discoaster nobilis* to *Fasciculithus involutus*) suggests a return to background rates of accumulation found in the pre-CIE interval, or ~0.047 m/k.y.

**4.5 Integrated marine microfossil changes (nannofossils, palynology, and foraminifera)**

Biostratigraphic data from the Anchor 3 core provide evidence for significant faunal changes and can be divided into three distinctive intervals: pre-, syn-, and post-CIE.

**4.5.1 Pre-CIE interval (late Paleocene, Selandian to Thanetian)**

Terrestrial palynomorphs comprise 30 to 70% of the palynomorphs in the pre-CIE interval with brief, periodic increases in
marine-dominant excursions (>85%) and a more prolonged marine-dominated interval (>60 to 80%) at around 60 m prior to the main CIE (Fig. 4). Marine dinocysts were present in most samples, but due to the dilution by terrigenous material and preservation effects, pre-CIE abundances in some samples are based on relatively low counts (<50 cysts). Thus, assemblages in these low abundance samples should be considered as rough estimates. Two notable marine dinocyst abundance peaks of near-shore lagoonal and shelfal taxa occur in cuttings samples 31790-31820 ft and 31820-31850 ft. These samples are
dominated by dinocysts in the Goniodomideae group (*Cordosphaeridium* complex, *Operculodinium centrocarpum*), the *Spiniferites* group, and initial increases and diversification of *Apectodinium* complex. These dinocyst peaks are followed by increasing dominance of spores and pollen, particularly increases in fern spores *Cicatricosisporites* and *Laevigatosporites* at 31660-31690 ft and 31690-31730 ft cutting intervals, and persistent *Pediastrum* and *Inaperaturapollenites haitus*.

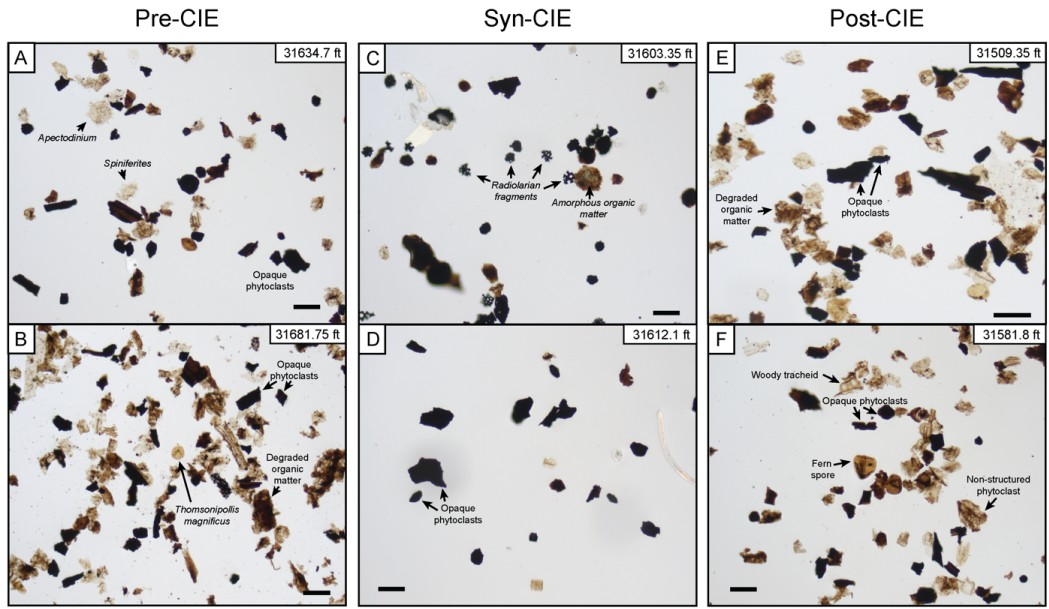

Figure 9. Photomicrographs of phytoclast and organic matter slides from core samples. Black scale bar is 50 μm. A) Mixed continental-marine palynofacies. Opaque to dark brown, equidimensional 'coaly' phytoclasts, light brown 'fresh' phytoclasts with visible plant structure, some degraded organic matter, pollen, spores, and dinocysts (note presence of *Apectodinium homomorphum*). B) Terrestrially derived, translucent structrured phytoclasts and non-structured light brown to darker brown organic material with microspores. Includes larger equidimensional 'coaly' black-brown opaque phytoclasts. C) Low abundance of organic material. Degraded, opaque 'coaly' phytoclasts, non-structured amorphous organic matter, and pyritized radiolarian fragments. Terrestrial pollen/spores and marine dinoflagellates are absent. D) Lower abundance of organic material, dominated by opaque, equidimensional inertinite and degraded phytoclasts. E) Mixed continental palynofacies with structured laths, equidimensional black-brown inertinite, and non-structured light brown 'fresh' phytoclasts. Pollen and spores present. Rare to no marine dinocysts. F) Mixed continental palynofacies with structured laths, equidimensional black-brown inertinite ('coaly' debris), and non-structured to structured brown, 'fresh' phytoclasts (cuticles, resinous fragments, tracheidal structures). Pollen and spores present. Rare to no marine dinocysts.

Kerogen preps with a low C/M ratio are abundant in terrestrially derived organic matter (OM) consisting of opaque to dark brown blocky phytoclasts, translucent phytoclasts (such as trachs and cuticle fragments), and pollen/spores (Figs. 9a-b). A low diversity assemblage of arenaceous foraminifera occurs in samples below the CIE interval and includes *Bathysiphon eocenica (=Nothia robusta?)*, *Hormosina velascoensis*, *Spiroplectammina spectabilis*, and *Trochamminoides* spp. (= *Trochamminoides variolarius*). Other forms of arborescent foraminifera include delicate fragments of small, and sometimes "glassy", forms of

*Bathysiphon*, which are probably varieties of *Nothia excelsa*. The assemblage of foraminifera in this interval, particularly *Spiroplectammina spectabilis*, suggests middle to the lower part of upper bathyal, or in the range from 500 to 700 m water depth (Van Morkhoven et al., 1986). In general, species attributable to *Nothia* are found in assemblages inhabiting water depths not shallower than the upper bathyal zone (Kaminski et al., 1996).

Radiolarian shells are present in low numbers (10-50 specimens) in many samples from the >63 μm fractions throughout the

pre-CIE interval, with only one sample at 31813.25 ft exceeding 10,000 specimens (Fig. 4). Preservation is poor to moderate with significant recrystallization and silica infilling of the frustule. Although preservation is poor, radiolarian shells were still



preserved well enough to distinguish between the Orders Spumellaria and Nassellaria; all assemblages are dominated by Spumellaria. In the modern Gulf of Mexico, significant radiolarian accumulations are found in sediments deposited on the continental slope (Pflum and Frericks, 1976).

Nannofossils are very sparse from 32370-33660 ft but markers can still be found with effort (Fig. 4). The Base of *Discoaster multiradiatus* is interpreted at 32233.25 ft which is uncharacteristically deeper than the Top of *Heliolithus kleinpellii* at 31843.35 ft. The Top of *Heliolithus riedelii* at 32340-32370 ft is of significance for designating oil and gas plays at approximately 57 million years, followed by the Base and Top respectively of *Toweius eminens* and a younger form of *Fasciculithus magnicordis* in the upper Selandian. Although *F. magnicordis* generally occurs within older portions of the
section, two variations are noted at shallower depths with slightly modified morphology of the central body that distinguish them from the holotypic *F. magnicordis*.

**4.5.2 Syn-CIE and recovery phase (late Paleocene/earliest Eocene)**

The relative abundance of marine dinocysts increases from 27% to 61% with counts of *Apectodinium* spp. increasing in relative abundances from 19% to 39% (between cutting sample 31660-31690 ft and core sample 31634.70 ft), ~3.5 m below the onset
of the CIE (Fig. 4). The increase in *Apectodinium* (particularly *A. homomorphum*) is immediately followed by two core samples (31612.1 ft and 31603.35 ft) within the main CIE that have low palynomorph abundances and a dramatic change in kerogen represented by opaque phytoclasts, fragments of pyritized radiolarians, and near absence of terrestrial and marine organic matter (Figs. 9c-d).

Sample residues in the size fraction >63 μm are generally absent of microfossils, with the exception of arenaceous foraminifera
and radiolaria. Arenaceous foraminifera characterize the low abundance foraminiferal assemblage, with indeterminate specimens and numerous fragments of small *Bathysiphon* spp. (=*Nothia excelsa*?). Radiolarians occur in both core and cutting samples, with highest abundances reaching as many as 10,000 specimens per sample at 31603.35 ft and 31612.10 ft.

Calcareous nannoplankton increase in abundance significantly across the CIE, with the most notable peaks in *Rhomboaster cuspis*, *Bromolithus elegans, Fasciculithus* species, *Toweius* species, and *Discoaster* species at 31603.35 ft. An acme of *D.*
*multiradiatus* the generic Eocene discoasters at 31600-31630 ft co-occurs with acmes in *Heliolithus* spp., *Fasciculithus* spp. (particularly, *F. tympaniformis*), and *Toweius* spp.

The CIE recovery phase is defined by C/M ratios of 0.43 and 0.84 with the return of translucent phytoclasts, near-shore and shelfal dinocysts (dominated by *Apectodinium* and *Spiniferites*), and terrestrial pollen/spores (Fig. 4). These changes, in association with a peak in freshwater *Pediastrum* and increases in *Bathysiphon* sp. and agglutinated forams, point to increasing
terrestrial sediment influences. Among nannofossils, this interval marks a decline in *Rhomboaster cuspis* and *Fasciculithus* species and maintains relatively high abundances of *Discoaster* spp. (mainly *D. multiradiatus* and *D. lenticularis*).

**4.5.3 Post-CIE (early Eocene, Ypresian)**

Following the negative CIE, terrestrial palynomorphs range from 65-98% of relative abundance with significant floral changeover documented in the terrestrial pollen/spores assemblages, most notable a decline in Juglandaceae (*Carya* and
*Momipites*-types) and persistence in normapolles, *Thomsonipollis magnificus* and *Nudopollis terminalis*. The marine palynological component documents an overall decline in species diversity and abundance, particularly in *Apectodinium* spp.




which declines to <5% in the post-CIE interval. Kerogen is dominated by mixed-opaque and translucent phytoclasts with abundant terrestrial spores and pollen indicating strong influence of terrigenous material to this locality (Figs. 6e-f).

Calcareous nannofossil excursion taxa decline in abundance above the CIE in both core and cutting samples and are replaced
by abundant fasciculiths and an acme of *D. multiradiatus* and *Toweius* spp. Foraminifera decrease with the exception of arenaceous fragments and small undifferentiated forms of *Bathysiphon* spp. Several specimens of moderately well-preserved *Nuttalides truempyi,* which are diagnostic of lower bathyal to abyssal water depths (1000 to 2000 m; Van morkhoven et al., 1986), were observed in the cuttings sample from 31570-31600 ft. Super abundant radiolarians, dominantly *Spumellaria*, occur at 31509.35 ft.

**4.6 Chemical index of alteration (CIA)**

In total, 71 mudstone and 48 sandstone samples were analyzed for major element geochemistry using XRF. Of the mudstone samples, 10 samples were excluded from CIA calculation due to the presence of calcite which can artificially depress calculated CIA values (Hessler et al., 2017). All but two of these calcite-bearing samples occur within the calcareous sub-type of LF-1 that occupies the upper portion of the main CIE (Fig. 7). High calcareous nannofossil recovery in this zone suggests a biogenic
source of calcite in these samples (Fig. 4). The remaining 61 mudstone samples yield an average CIA of 66. However, a step-like shift in CIA occurs ~4 m above the CIE onset (between samples 117 and 116). Below this shift, 88% of samples have CIA higher than average and above this shift 77% of samples have CIA lower than average (Fig. 7).

Of the sandstone samples analyzed, five samples exhibited anomalous cementation as confirmed by XRD analysis and were excluded from CIA calculation (Fig. 7; Figure S1). The average CIA for the remaining 43 sandstone samples is 55.8. Similar
to the mudstone samples, CIA values from sandstone samples exhibit a step-like shift towards lower values upwards in the section. However, the shift occurs within the CIE recovery (between samples 97 and 80), ~9 to 10 m above the CIE onset, ~5-6 m higher than the corresponding shift in mudstone CIA values (Fig. 7). Below this shift, 95% of samples yield a CIA that is higher than average and above the shift 90% of samples have a CIA that is lower than the average value (Fig. 7).

**5 Discussion**

**5.1 Sedimentologic expression of the PETM, deep-water Gulf of Mexico**

**5.1.1 Lithology**

The PETM is manifested in four distinctive lithologic intervals (A-D; Fig. 5). Unit A comprises the lower 1.7-2.3 m of the CIE and is characterized by similar sedimentary facies and organic content as the underlying 66 m, suggesting a lag between the onset of the PETM and its associated lithologic response (Fig. 5; see also section 5.1.4). Unit B extends from the top of Unit A
to ~4.5 m above the onset of the CIE and represents a transition to organic-lean claystone that is conspicuously poor in palynomorphs, silt, and sand relative to below (Fig. 4). Although the PETM CIE was not identified in the wells studied by Cunningham et al. (2022), they reported that other GOM PETM sections were dominantly light gray shale, with lesser siltstone and marlstone, suggesting that the record preserved in the Anchor 3 core may be broadly representative of lithologic change during the PETM over the greater Wilcox deep-sea fan.



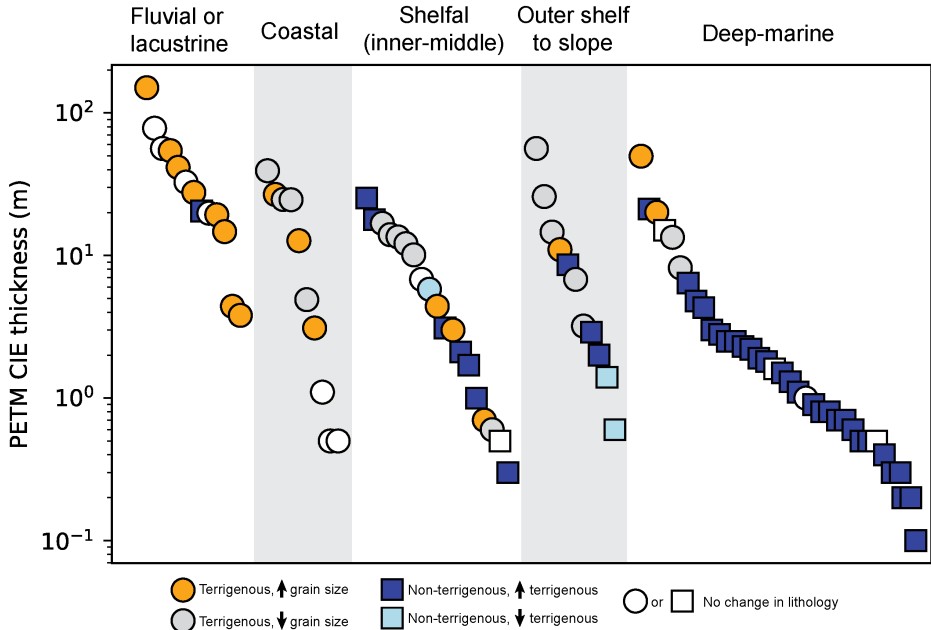

Figure 10. PETM CIE thickness plotted in descending order by depositional environment (see Table S1). Marker shape corresponds to dominant sediment source (terrigenous vs non-terrigenous), and marker color corresponds to lithologic change from before to during the PETM. Localities where lithologic change is uncertain are not shown.


Many other distal marine PETM sections (shelfal, slope, deep-sea) that are connected with terrigenous sediment sources also show an increase in clay deposition (i.e., decrease in grain size) during the PETM CIE, including in nearshore shelfal sediments in Spitsbergen (Harding et al., 2011), inner-middle neritic sediments of New Jersey (Self-Trail et al., 2012), outer shelf to slope sediments of central California (John et al., 2008), and outer neritic to deep-marine sediments of the North Sea Basin (Kender

et al., 2012; Schoon et al., 2015) (Fig. 10). Most open-marine, non-terrigenous PETM localities also show an increase in clay relative to carbonate during the PETM that is thought to reflect a combination of carbonate dissolution due to ocean acidification and enhanced rates of terrestrial sediment delivery to the global ocean (Zachos et al., 2005; Sluijs et al., 2008a). Indeed, the few PETM sections that display no change or an increase in carbonate content are considered anomalous and have been ascribed to either extreme increases in primary productivity (e.g., Bolle et al., 2000) or distance from terrestrial landmasses (e.g., Jiang

and Wise, 2009; Robinson et al., 2011). Thus, lithologic change during the PETM within the Wilcox deep-sea fan shares a similar pattern to many other marine PETM localities that experienced increased accumulation rates of terrestrial clay (Fig. 10; see also Section 5.4).

Unit C lies within the main CIE and reflects a lithologic change to marlstone with an associated increase in $CaCO_3$ relative to underlying claystone and lesser siltstone in Unit B (Figs. 5 and 7). High abundances of calcareous nannofossils in Unit C

suggests a biogenic origin of the calcite (Fig. 4). Concentrations of $CaCO_3$ are generally very low in the siliciclastic-dominated Wilcox Group, and thus the marlstone of Unit C is anomalous. An initial rapid decrease and subsequent recovery in $CaCO_3$ is a common characteristic of open-marine PETM localities that record ocean acidification and shoaling of the calcite



compensation depth during the early stages of the PETM (e.g., Zachos et al., 2005). Thus, ocean acidification in the Gulf of Mexico during the early PETM may explain the lack of $CaCO_3$ in Unit B relative to Unit C (Canudo et al., 1995; Bralower et al., 1997; Colosimo et al., 2005; Zachos et al., 2005; Raffi et al., 2009; Self-Trail et al., 2012; Penman et al., 2014; Gutjahr et al., 2017). Assuming an average sedimentation rate of 0.063-0.068 m/k.y. for the ~13 m thick PETM CIE (~200 k.y.; Westerhold et al., 2018), we calculate deposition of $CaCO_3$-rich mudstone to postdate the onset of the PETM by 70 ± 7.5 k.y., an estimate that is in good agreement with deeper Ocean Drilling Program sites 1262 and 1267 from the South Atlantic that show $CaCO_3$ increases beginning ~60-70 k.y. post-CIE onset (Zachos et al., 2005). The excursion taxa *Rhomboaster* spp., which have highest abundances in the main CIE (Fig. 4), have also previously been associated with low pH conditions (Self-Trail et al., 2012).

Unit D, which is interpreted to represent the CIE recovery, exhibits a return to thinly bedded mudstone and siltstone (LF-2) that is similar to pre- and post-CIE fine-grained deposits (Fig. 7). Unit D records a resumption of silt delivery to the basin that shows a corresponding change in kerogen type with an increase in well-preserved, structured terrigenous phytoclasts and pollen/spores mixed with shelfal to neritic dinocysts that is often indicative of turbiditic muds (Tyson, 1995) (Figs. 9e-f). Although lowermost Eocene (post-CIE) lithofacies are generally similar to those in the uppermost Paleocene (pre-CIE) (Fig. 7), there is some indication of a persistent change following the PETM. Gamma-ray log motifs change from coarsening-upwards to fining-upwards (Fig. 7), suggesting a possible change in deep-water stratigraphic architecture. This observation, in combination with the appearance of extensive mudclast conglomerate in associated with thickly bedded sandstone in the lower Eocene, may suggest an increase in deep-water channelized facies across the Paleocene-Eocene boundary (Figs. 7 and 11). Terrestrial spores and pollen become dominant over marine dinocysts following the PETM, indicating a corresponding increase in terrigenous sediment flux relative to late Paleocene time (Fig. 3). Thus, the Anchor 3 core may reflect an increase in terrigenous sediment supply to the deep-water GOM that persisted long after the PETM, similar to the inferences of some other studies (Foreman et al., 2012; Borneman et al., 2014).

### 5.1.2 Organic content

Units B and C of the main CIE display a ~6-fold decrease in TOC relative to the pre-CIE and post-CIE shale average of ~0.9% (Fig. 4). The pronounced decrease in TOC in the Anchor 3 core during the PETM CIE is anomalous relative to many other marine PETM sites that exhibit enhanced rates of organic carbon burial (e.g., Gavrilov et al., 2003; Sluijs et al., 2008a; Schoon et al., 2015). However, nearshore shelfal sediments of the Van Mijenfjorden Group in Spitsbergen display an approximately 2-fold drop in TOC from ~3% before the PETM to ~1.5% during and after the CIE (Harding et al., 2011), a similar albeit less pronounced decline in TOC as compared to the Anchor 3 core.



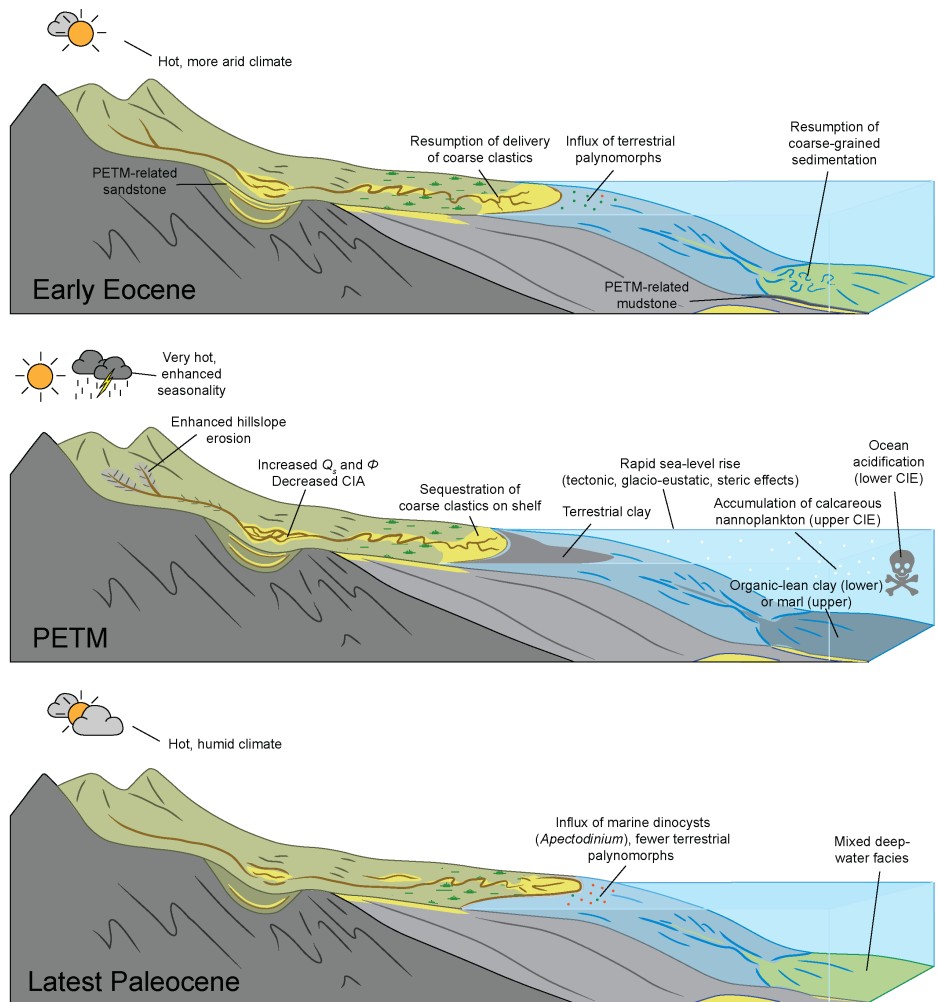

Figure 11. Interpreted patterns of sediment delivery from land to deep-sea before, during, and after the PETM (modified from Romans and Graham, 2013; see Discussion text).

Decreased TOC within Units B and C of the main CIE may reflect an influx of TOC-poor terrigenous clay into the Gulf of Mexico relative to the flux of marine organic carbon. Decreased paleosol TOC within the western United States has been linked

with enhanced oxidation of organic matter within terrestrial environments during the PETM (Wing et al., 2003; Carmichael et al., 2017). Visual examination of kerogen slides from core and cuttings samples indicates the presence of degraded, likely oxidized, terrigenous carbon that includes black structured to non-structured carbonized debris and inertinite (Figs. 9c-d). The low TOC within claystone and marl of Units B and C is thus suggestive of extreme siliciclastic dilution relative to primary marine productivity in the Wilcox deep-sea fan. Thus, unlike marine shales that are thought to reflect condensed sections

formed by sediment starvation to the basin, the clay-rich shale and marlstone of the PETM likely reflects a pronounced increase in terrigenous clay delivered to the deep-sea (Fig. 11).





The organic content of Unit D fluctuates between pre-CIE and main CIE values (Fig. 5), perhaps reflecting delivery of organic carbon from different sources (e.g., Aze et al., 2014), varying rates of sediment accumulation relative to marine productivity, and/or delivery of less oxidized organic matter to the basin. Carbonaceous organic debris was noted frequently in visual inspection of samples from Unit D relative to Units B and C of the main CIE (Fig. 5). Thus, the CIE recovery recorded in Unit D is interpreted to record a transition low TOC, clay-rich deposition during the main PETM CIE to more typical Wilcox Group lithologies that were deposited post-CIE.

### 5.1.3 Chemical weathering

The CIA, a proxy for silicate mineral weathering, displays a step change within the PETM from higher values in the uppermost Paleocene to lower values in the lowermost Eocene (Fig. 7). This pattern suggests a transition to deposition of less weathered sediment during and after the PETM, with the shift occurring earlier in mudstone ~3.8-4.2 m above (~62 ± 5 k.y. after) the CIE and the shift in sandstone occurring ~9.5-9.9 m above (~149 ± 5 k.y. after) the CIE (Table S2).

The shift to lower CIA values within and after the PETM is anomalous relative to many other PETM sections that have been interpreted to record enhanced chemical weathering during the PETM (e.g., Ravizza et al., 2001) and, more generally, early Eocene time relative to late Paleocene time (Hessler et al., 2017). For instance, CIA values drop only briefly during the PETM onset before increasing during the main CIE and recovery in outer shelf deposits of Spitsbergen (Wieczorek et al., 2013). Similarly, Chen et al. (2016) show a gradual increase in CIA during the PETM in lacustrine sediments of central China. Increased CIA values in the Tethys during deposition of highly organic sediments have been associated with erosion of highly weathered soils (Dickson et al., 2014).

The shift to lower CIA values within and after the PETM in the Anchor 3 core may be explained by enhanced erosional denudation within central North America. For instance, Foreman et al. (2012) suggested a system-wide cleansing of catchment colluvium and an associated increase in bedrock erosion during the PETM, associated both with vegetation change (e.g., Wing et al., 2005) and an intensification of the hydraulic cycle. CIA values increase upwards in soil profiles (Nesbitt and Young, 1982), such that erosion into deeper levels of soil profiles may produce CIA values that become closer to fresh bedrock (CIA of ~45 for average continental crust; McLennan, 1993). The negative relationship between denudation rate and CIA of suspended sediment in modern rivers (McLennan, 1993) is consistent with the inference that an increase in denudation rate during the PETM could result in a lowering of CIA values of terrigenous sediment reaching the GOM.

### 5.1.4 Temporal lags in PETM signals

Although the global negative CIE associated with the PETM was synchronous with warming (Zeebe et al., 2016), lithologic and geochemical changes in the Anchor 3 core lag behind the onset of the PETM (Figs. 5 and 7). Following the approach of Duller et al. (2019), we estimate an approximate temporal lag of 31 ± 6.2 k.y. between the onset of the PETM and the abrupt reduction in grain size that marks the transition between Units A and B. A similar delay in the sedimentary response to the PETM was noted by Duller et al. (2019) who estimated temporal lags of 16.5 ± 7.5 k.y and 16.5 ± 1.5 k.y for terrestrial and deep-marine sites, respectively, in the Spanish Pyrenees. In proximal, fluvial environments the temporal lag was associated with a delay in coarse-grained (conglomeratic) sedimentation, whereas in deep-marine environments the lag was associated with a delay in increased mass accumulation rates of terrestrial clay, which replaced limestone as the dominant lithology during the PETM (Schmitz et al., 2001; Duller et al., 2019). John et al. (2008) inferred a ca. 20 k.y. delay in increased mass





accumulation of carbon and carbonate in PETM sites in central California and the New Jersey shelf. A deep-marine PETM section within the North Sea also displayed a lag of ~1-2 m between the onset of the CIE and a transition to laminated mudstone (Kender et al., 2012), although the lack of definition of the CIE recovery precludes calculation of a lag time at this locality.

Theoretically, PETM-related signal lag times should depend on a number of factors summarized by Tofelde et al. (2021) that include (1) the sediment size fraction that carries the signal, with signals held in the finger-grained fraction propagating downstream more quickly than those held in coaster sediment fractions, (2) the relative proximal-to-distal position of the site of deposition, where proximal sites record changes sooner than distal sites, and (3) for marine sections, the degree of connectivity with upstream fluvial systems (e.g., Bernhardt et al., 2017). The delay in shut-off of sand and silt to the Wilcox deep-sea fan is likely related to rapid sea-level rise during the onset of the PETM that culminated in maximum flooding during the CIE peak, ~10-13 k.y. after the onset (Sluijs et al. 2008b; Harding et al., 2011). Deposition of sand and silt in Unit A thus persisted well after maximum flooding had been obtained. Continued coarse-grained sedimentation may reflect mobilization of sediment already staged within the submarine portion of the sediment routing system that continued to move basinward, despite a shut-off in supply of coarser grained sediment.

Similarly, the delayed arrival of abundant terrestrial clay to the Wilcox deep-sea fan likely reflects the timescale over which the PETM signal of increased terrigenous sediment supply was transferred to the distal portion of the Wilcox sediment routing system. Numerical and analog experiments suggests that changes in climatic boundary conditions (e.g., an increase in precipitation) are manifested rapidly as a change in the amount of sediment exported from erosional landscapes (Bonnet and Crave, 2003; Armitage et al., 2011), such that the signal onset time (*sensu* Tofelde et al., 2021) is close to zero. Thus the 31 ± 6.2 k.y. lag between CIE onset and lithologic response likely represents the time for the signal of increased sediment generation to transfer 100s of km to the site of deposition in the Wilcox sediment routing system, rather than reflecting a delay in landscape response to the PETM.

The step-change in shale and sandstone CIA also lags behind both the onset of the CIE and associated change in lithology. We speculate that the delayed arrival of low-CIA shale (~62 ± 5 k.y. post-CIE) relative to the transition to clay-rich shale (~31 ± 6.2 k.y.) may reflect the timescales of erosional denudation in the hinterland. Initial erosion during the early PETM would first remove existing soil and colluvium before exposing less-weathered soil and bedrock. Thus the ~30 k.y. offset in signal arrival times within the mud sediment fraction may reflect progressive erosional denudation of the North American hinterland during the PETM (e.g., Foreman et al., 2012).

The ~90 k.y. offset in signal arrival times between decreases in shale and sandstone CIA (62 ± 5 k.y. vs ~149 ± 5 k.y., respectively) may reflect different rates of sedimentary signal propagation associated with the hydraulic grain size fraction in which they are carried as signals are carried more quickly in fine-grained vs coarse-grained fractions due to the relative speed in which clay and fine silt are transported within sedimentary systems (Ganti et al., 2014; Tofelde et al., 2021).

**5.2. Biotic expression Paleocene-Eocene boundary interval, deep-water Gulf of Mexico**

The marine biotic response to the climatic warming event of the PETM is well documented in localities world-wide (e.g., Thomas and Shakleton, 1996; Kahn and Aubry, 2004; Crouch et al., 2001; Sluijs et al, 2007a; Denison, 2021) and in the GOM, where it is reflected in the assemblage changes or turnover in specific groups of single-celled planktonic organisms, most notably marine dinoflagellates, benthic foraminifera, and calcareous nannoplankton. Interpreting the deep-water GOM record



at any single location may be complicated by a limited recovery of calcareous microfossils but distinct trends noted in wells across the basin and in the Anchor 3 core suggest multiple factors played a role in calcareous and organic fossil abundances. Published data from other GOM PETM localities is limited because of relatively few borehole penetrations and the often proprietary nature of company data, making this record an important addition to understanding both the global and regional variability of this event. The Anchor 3 core provides an opportunity to better understand the complex relationship between deep-water depositional facies and processes, climatic changes, and preservation of fossil abundances in a bathyal PETM

section.

The diversity and distinctiveness of faunal patterns in dinocyst data at Anchor 3 record similar trends in common late Paleogene taxa (including *Apectodinium* group) documented in onshore eastern GOM localities (e.g., Denison, 2021; Slujis et al., 2014, Harrington and Kemp, 2001) and from other localities world-wide (e.g., Crouch et al., 2001; Steurbaut et al., 2003, Sluijs et al., 2007a; Sluijs and Brinkhuis, 2009; Smith et al., 2021; Frieling and Sluijs, 2018). Short-lived taxa increases prior to the CIE

occur in *Cordosphaeridium* complex, *Eocladopyxis peniculata* (and other epicystal Goniodomideae taxa), *Adnatospaheridium multipinosum*, *Operculodinium* spp., *Areoligera* group, and *Spiniferites* spp., indicating that multiple factors of climatic/paleoenvironmental change (such as temperature, sea level, salinity, and nutrient availability) impacted species dominance at this deep-water locality. The peak intervals of mixed lagoonal/nearshore to neritic taxa correlate well to depositional lithofacies LF-2 and include abundant pollen and spores (such as bissacate and swamp taxa) and opaque to

structured phytoclasts. This mixed terrestrial to marine assemblage points to long transport distance from the initial environments of deposition. Furthermore, the presence of lagoonal dinocysts, particularly *Eocladopyxis* and *Apectodinium* that thrive in brackish water environments, infer periods of peak continental runoff and enhanced nutrient availability prior to the CIE (Sluijs and Brinkhuis, 2009; Sluijs et al., 2014).

A unique and interesting biotic response documented in low, mid-, and high latitude localities is a global acme of *Apectodinium*

spp. that precedes and is contemporaneous with the late Paleocene negative CIE (Denison, 2021). The increase in *Apectodinium* abundance has primarily been explained by increasing temperature in high-latitude sites across the PETM (e.g., Crouch et al., 2001; Frieling et al., 2014). In addition to being thermophilic, *Apectodinium* distribution and abundance may be influenced by other environmental factors such as salinity, ocean productivity, and ocean stratification (Sluijis and Brinkhuis, 2009). Fluctuations in *Apectodinium* abundance from the Tethys (Crouch et al., 2003a, b) and Spitsbergen (Harding et al., 2011)

suggests a linkage to increases in continental sediment delivery and sea level. Additionally, select mid-to-low latitude sites have documented that the *Apectodinium* acme precedes the CIE, suggesting that a latitudinal temperature gradient and additional environmental factors may have contributed to higher *Apectodinium* abundances (Sluijs et al., 2007b, van Roij, 2009).

The consistent presence of *Apectodinium* in pre-CIE samples (~120 m prior) infers optimum environmental conditions for the

establishment of this taxon in the late Paleocene GOM. At Anchor 3, not only is *Apectodinium* consistently present in pre-CIE samples, but also the onset of the acme of *Apectodinium* occurs in the sample just prior (~3.5 m) to the main CIE at 31634.7 ft (Fig 4). The sample of maximum *Apectodinium* abundances occurs in the muddy siltstone of LF-2 and includes the presence of neritic taxon, *Spiniferites* group and *Adnatospaheridium multispinosum*; small numbers of *Areoligera* complex; terrestrially-derived kerogen, pollen and spores; and lacks previously observed *Pediastrum* and Goniodomideae taxa, suggesting an overall

more distal influence either from sea level rise, decline in fluvial influence, or both.





A low palynomorph abundance interval occurs during the main CIE and corresponds with a low TOC, clay-rich interval of LF-1 and an increase of radiolarians and calcareous nannofossil excursion assemblage, the *Rhomboaster* spp. - *Discoaster* spp. association (RD; Kahn and Aubry, 2004). The RD is a unique assemblage of species of two genera that spans the CIE and along with the *Apectodinium* acme, is used as a proxy for the CIE. The discoasters (*D. araneus* and *D. anartios*) are tightly

constrained to the isotopic excursion, whereas the rhomboasters often range into the uppermost Paleocene, thus constraining the PETM in conjunction with the isotopic curve.

The abrupt decline in palynomorph abundance, including *Apectodinium*, suggests an interplay of multiple factors including the rise of sea level, decline in coarser-grained sediment influx, and deoxygenation just prior to and during the PETM. Similar observations have been documented in Harrell Core, onshore Mississippi (Slujis et al., 2014) and Site M007, offshore Mexico

(Smith et al., 2021). The kerogen in this organic poor interval is dominated by opaque phytoclasts and pyritized radiolarian fragments (Figs. 9c-d). High amounts of opaque phytoclasts result from oxidation and are present in settings that are either proximal to terrestrial environments or that occur in distal settings suggestive of long transport time (Tyson, 1995). Our interpretation at the Anchor 3 bathyal locality precludes the former, as the oxidized material was transported with minimal phytoclasts or palynomorphs within this clay-rich, organic-poor interval. Similarly, the Spitsbergen locality demonstrates that

in a sediment starved environment, *Apectodinium* and other marine dinocysts were absent in the main CIE (Harding et al., 2011). In the LF-2 of the recovery phase, *Apectodonium* spp. and other marine dinocysts and terrestrial palynomorphs return further suggesting a facies-controlled decline of palynomorphs during the peak CIE at the Anchor 3 locality.

Radiolarian oozes are abundant in lower to middle Eocene intervals throughout low latitude sections, including many wells from the USA GOM. While previous studies have documented radiolarian rich intervals during the PETM in cuttings samples

(e.g. Cunningham et al., 2022), we report radiolarian-dominated intervals from core during the CIE. In general, radiolaria are abundant in modern slope sediments where dissolved silica concentrations are relatively high to favor shell production, export, and preservation. Enhanced dissolved silica concentrations may have resulted either from upwelling within the basin or terrestrial runoff of silica from up-dip catchments that drained into the GOM. Because palynomorph assemblages contain abundant terrestrial palynomorphs and lack indicators of enhanced productivity within the same samples, we consider enhanced

runoff to be the more significant factor.

Similar to dinocysts, calcareous nannofossils underwent temporary but significant assemblage changes during the PETM. These variations provide an additional tool to track the biotic and climatic response across this boundary, specifically within the genera *Rhomboaster*, *Discoaster* and *Fasciculithus* (e.g., Bralower, 2002; Kahn and Aubry, 2004; Agnini et al., 2006; Self-Trail et al., 2012). The abrupt increase in these PETM excursion taxa may be linked to rise in surface water temperatures as

previous research points to these species as being adapted to warm and oligotrophic conditions (e.g., Bralower, 2002).

Though influxes of *Toweius* spp. and *Discoaster* spp. are sometimes used as indicators of inhospitable water column conditions (Self-Trail et al. 2012), this does not appear to be the case at the Anchor 3 locality because of the abundances observed across multiple genera. Fasciculiths and discoasters are also robust forms that are less prone to dissolution than others. The few occurrences of *D. araneus* (and absence of *D. anartios*) in that section could be due to water column dynamics or a sampling

artefact. The fact that the small and thick *R. cuspis* is the only member of the RD to be preserved, alongside the more robust Fasciculiths, may indicate that the more delicate forms of Rhomboaster (e.g., *R. calciptrapa*) were not preserved.

**5.3 Paleobathymetry and bottom water oxygen**



Water depths over the cored interval range from the lower part of the upper bathyal to the lower bathyal zone, or 500 to 2000 m. Several species of observed foraminifera, such as *Nothia* spp., *Spiroplectammina spectabilis*, *Trochamminoides variolarius*, and *Nuttalides truempyi*, are diagnostic indicators. Additionally, radiolarian abundances occur in the core along with lower diversity agglutinated foraminiferal assemblages. Together, these observations suggest bathyal (likely mid to lower) water depths given typical hypsographic depth ranges for these levels in the GOM.

Although the lithologic response to the PETM was delayed, there was an immediate decrease in bioturbation that persisted for ~2.5-3 m above the CIE onset followed by a gradual increase in bioturbation into Unit C (Fig. 7). Decreases in bioturbation in the lower PETM CIE have been noted in several other marine PETM localities (Bralower et al., 1997; Nicolo et al., 2010; Schulte et al., 2011; Mutterlose et al., 2017). For example, Nicolo et al. (2010) documented an abrupt drop in trace fossil abundance coincident with the onset of the PETM CIE at Mary and Dee streams, New Zealand, with a recovery to pre-CIE levels approximately mid-way through the PETM. Deficiency in seafloor oxygenation and associated stress to benthic organisms during the early CIE has been invoked as a likely explanation for the decrease in bioturbation during the early PETM (Nicolo et al., 2010; Shulte et al., 2011). For example, Bralower et al. (1997) associated faint laminations in claystone deposited during the main phase of the PETM CIE to reflect decreased bioturbation and dysoxic bottom waters. Decreases in oxygen levels have been inferred for other deep Atlantic PETM localities on the basis of Mn and U enrichment factor and I/Ca proxies (Chun et al., 2010; Pälike et al., 2014). The Anchor 3 core suggests that deoxygenation also occurred in the Gulf of Mexico during the early phases of the CIE.

The lack of benthic foraminifera in sediments from the CIE interval is further evidence of low benthic oxygen concentrations during the PETM. A benthic foraminiferal extinction is observed in many classic PETM sections and is largely considered a consequence of significant reduction of oxygen availability (Kennett and Stott, 1991; Sluijs et al., 2014). Unlike other localities spanning late Paleocene to early Eocene time, we note very low diversity foraminiferal assemblages not just during the CIE but throughout the entire core. A characteristic relatively unique to the GOM Wilcox Group are agglutinated benthic foraminifera dominated by epifaunal arborescent foraminifera and sparse occurrences of planktonic or calcareous foraminifera. A lower diversity and dominance of just a few species are indicators of dysoxic to perennially anoxic environments in modern silled basins (e.g. Bernhard et al, 1997), but these settings are also characterized by infaunal species. In contrast to typical low oxygen assemblages, the benthic foraminifera reported here are primarily epifaunal. A singular factor does not explain the faunal trends noted in the Anchor 3 core. A combination of factors, such as relatively high sedimentation rates, reduced bottom water oxygen concentrations, and paleobathymetry at or below the CCD would best explain the observations from the benthic foraminifera.

**5.4 Grain size partitioning from land to ocean during the PETM**

Although rates of terrigenous sediment supply to the world's oceans are largely inferred to have increased during the PETM (Crouch et al., 2003b; John et al., 2008; Sluijs et al., 2008a, 2008b, 2011; Aze et al., 2014; Bornemann et al., 2014; Carmichael et al., 2017; Dunkley Jones et al., 2018; Duller et al., 2019), a comparison of lithologic change during the PETM from proximal to distal depositional settings reveals opposite trends in grain size change in terrestrial versus marine settings (Fig. 10). A majority of proximal terrestrial PETM localities record an increase in grain size, whereas terrigenous-sourced marine PETM localities most often record a decrease in grain size (Fig. 10; Table S1). This dichotomy is well illustrated in the Wilcox sediment routing system, where proximal fluvial sections illustrate increases in coarse-grained sedimentation during the PETM





(Foreman et al., 2012; Foreman, 2014; Dechesne et al., 2020) whereas the distal deep-sea fan records an opposite change to
        deposition of finer-grained sediments. Coastal PETM sections, deposited at the interface between terrestrial and marine
        environments, show an approximately even split between an increase and a decrease in siliciclastic grain size (Fig. 10; Table
        S1).

        These observations suggest pronounced partitioning of grain size during the PETM, with coarser-grained sediment sequestered
in coastal or shelfal environments and finer-grained sediment (clay) distributed widely in the deep-sea. This partitioning is
        likely a consequence of two competing changes in boundary conditions that occurred synchronously during the PETM: (1) an
        increase in the supply of both coarse- and fine-grained grained sediment and (2) sea-level rise. If the PETM were manifested
        as an increase in sediment supply without a corresponding rise in sea level, it is likely that connectivity between terrestrial and
        deep-marine sediment routing systems would be maintained such that an increase in sand and silt in terrigenous-sourced PETM
localities would be observed, albeit with a possible lag between the arrival of fine- and coarse-grain size fractions (Tofelde et
        al., 2021). Instead, it appears that terrestrial and marine segments of many sediment routing systems became disconnected
        during the PETM as a consequence of sea level rise (Fig. 11). The ability for terrestrial clay to become widely dispersed in the
        ocean despite high sea level is evidenced by a near-uniform global increase in clay deposition within normally non-terrigenous
        marine PETM sections (Fig. 10; Khozyem et al., 2013, Giusberti et al., 2007; Canudo et al., 1995; Crouch et al., 2003a;
Bornemann et al., 2014).

        A deep-water PETM section within the Spanish Pyrenean Basin provides an exception to the above statements, where the
        PETM is manifested as an increase in siliciclastic grain size within deep-water channelized deposits (Pujalte et al., 2015). The
        Orio section shows a change from tabular, massive, mudstone-capped sandstone beds to amalgamated, planar-laminated,
        coarse-grained deposits with an axial deep-water channel belt, interpreted to reflect a change from sediment-gravity flow to
hyperpycnal flow dominated sedimentation (Pujalte et al., 2015). The difference in response between the Spanish Pyrenean
        Basin and the deep-water GOM may reflect differences in connectivity between terrestrial and marine segments of their
        respective sediment routing systems. For example, Bernhardt et al. (2015) noted varying degrees of connectivity between
        fluvial and deep-sea segments of the Biobío canyon system of Chile that related to the position of submarine canyon-heads
        with respect to fluvial and longshore sediment delivery systems. Margins with narrow shelves (e.g., southern California;
Covault et al., 2007) or where submarine canyons incise across the shelf to connect directly fluvial systems (e.g., the Congo
        River and associated deep-sea fan; Vangrieshiem et al., 2009) are known to maintain connectivity between land and deep-sea
        regardless of sea level state. Such settings are thus more likely to record an increase in coarse-grained sedimentation in the
        deep-sea during the PETM versus margins with broad shelves in which sea level rise would more effectively sequester coarse-
        grained sediment in coastal and shelfal settings.

**5.5 Evaluation of the Gulf of Mexico closed gateway hypothesis**

        Several studies have argued that the GOM became isolated from the Atlantic Ocean due to collision of the Cuban arc with the
        Yucatán, Bahamas, and Florida carbonate platforms during late Paleocene-early Eocene time, and that the GOM underwent
        approximately 1-2 km of drawdown during or just prior to the Paleocene-Eocene boundary (Rosenfeld and Pindell, 2003;
        Cossey et al., 2019; Pindell and Cossey, 2020; Rosenfeld, 2020; Cossey et al, 2021). Proponents of the closed gateway
hypothesis have suggested that release of insert methane hydrates during sea level fall might explain the large release of





isotopically light carbon and associated global warming that characterizes the PETM (Rosenfeld and Pindell, 2003; Cossey et al., 2016, 2021).

Several observations from the Anchor 3 core are inconsistent with the closed gateway hypothesis and instead support the interpretation that the GOM remained connected with the Atlantic Ocean (Cunningham et al., 2022). The Anchor 3 core does
not show evidence for oceanic drawdown or unconformity occurrence during or immediately prior to the PETM. Marine dinocysts are dominant over terrestrial spores and pollen in the Wilcox 1B interval preceding the Paleocene-Eocene boundary (Fig. 3), suggesting a relative lack of terrestrial input to the basin. Although the drawdown hypothesis invokes widespread paleo-canyon formation and increased clastic delivery to the deep GOM during or immediately preceding the PETM, the Anchor 3 core indicates the opposite: a reduction of grain size during the main PETM CIE that is interpreted to reflect sea level
rise, not sea level fall. If a drawdown of water occurred over the PETM interval, a shift from deep-water to shallow benthic assemblages would be expected. Our data do not show that, but instead indicate deep-water agglutinated foraminifera immediately below and above the CIE. The decline in terrestrial phytoclasts and spores/pollen, increase in opaque phytoclasts, and marine dinocyst assemblages with peaks in *Spiniferites* complex and *Areoligera* abundances, corroborate the foraminifera record prior to the CIE and are consistent with an overall trend in sea level rise (Sluijis et al., 2008a; Sluijis and Brinkhius,
2009). Furthermore, the deep-water GOM record across the Paleocene-Eocene boundary lacks evidence for deposition of evaporites or highly organic units that have characterized isolation and drawdown episodes of the Mediterranean and Black seas (Cunningham et al., 2022).

The PETM record from the GOM shares several features with other deep Atlantic Ocean sites, including an extinction in benthic foraminifera, a shift from $CaCO_3$-poor to -enriched sediment, and a decrease in bioturbation during the lower PETM CIE (e.g.,
Kennett and Stott, 1991; Zachos et al., 2005). These commonalities suggest similar oceanographic changes between the GOM and Atlantic Ocean during the PETM that would be best explained by these ocean basins being connected during the latest Paleocene to earliest Eocene time. Thus, our analysis casts doubt on the hypothesis that methane hydrate release as a consequence of GOM drawdown was a causal mechanism for the PETM.

## 6 Conclusions

The Anchor 3 core contains one of the few known PETM records from a deep-sea fan system sourced from a continental-scale drainage, providing insight into the effects of this hyperthermal on the landscapes of central North America and within the GOM ocean basin itself. The PETM is manifested by a $\sim$-2‰ shift in bulk organic $\delta^{13}C$ and a pronounced decrease in the supply of sand and silt after a lag of ~2 m (approximately 31 ± 6.2 k.y.), concurrent with an influx of terrigenous clay that is notably poor in organic carbon. Comparison of the Anchor 3 core with a global compilation of PETM localities suggests that
the lithologic manifestation of the PETM in the deep-water GOM was caused by the interplay of two factors: (1) increased erosional denudation of central North America that caused an increase in the supply of both coarse and fine-grained sediment to the coastal ocean and (2) rising sea level that sequestered coarser-grained sediment near the coast but allowed terrigenous clay to reach deep-sea environments. A stepped increase in the chemical index of alteration for both shale and sandstone during the PETM supports the inference of intensified erosional denudation that outpaced elevated rates of chemical weathering
associated with high temperatures.

The record of the PETM at the Anchor 3 well indicates that global climatic changes also had a clear influence on calcareous, siliceous, and organic-walled microfossil assemblages in the GOM. The latest Paleocene interval was characterized by

increasing abundances of *Apectodinium* spp. with mixed coastal to neritic dinocysts assemblages, shifts in the abundance of terrestrially derived kerogen, and fluctuations in radiolarian abundance that together suggest variability in sediment supply and

nutrient enrichment. The *Apectodinium* dominance and acme event prior to the main CIE are interpreted as a response to not only the global temperature change but also enhanced runoff and increased delivery of nutrients to the surface water in the GOM. The sudden absence of these allochthonous palynomorphs and organics during the PETM can be explained by sea level rise and sequestering of coaster-grained sediment that typically delivered these palynomorph assemblages basinward. Abundant radiolarians and the lack of calcareous microfossils, except for dissolution resistant nannofossils, suggest rapid sediment

accumulation rates and a shoaling of the CCD. An anomalous lack of bioturbation and a benthic foraminiferal extinction following the PETM onset are consistent with deoxygenation of GOM bottom-waters. The transition from $CaCO_3$-poor to $CaCO_3$-rich shale suggests ocean acidification during the early phases of the PETM. Similar observations have been made in other deep Atlantic PETM localities, suggesting connection of the GOM with the Atlantic Ocean during latest Paleocene-earliest Eocene time.

The Anchor 3 locality provides the first integrated lithologic, geochemical, and biostratigraphic record of the PETM in the deep-water GOM and is an essential record in further understanding the variability of the biotic response globally. This study reinforces the importance of multidisciplinary characterization of the PETM for understanding how climate-driven sedimentary signals are manifested within the distal portions of large sediment routing systems. The effects of oceanographic changes during the PETM onset (e.g., a decrease in bioturbation, benthic foraminiferal extinction) were immediate, while lithologic and

geochemical changes lagged by ca. 30 to 150 k.y. These results highlight both the immediate and long-lived consequences of global warming, even within the most distal portions of large sediment routing systems.

**Code/Data availability**

Data that accompany this article are available in Sharman et al. (2022).

**Author contribution**

G. Sharman and E. Szymanski initiated the project, framed the study, and gained partner approval for access to geologic materials that are the basis for all technical analytical work. G. Sharman described core samples and collected XRD data. R. Hackworth, A. Kahn, and L. Febo collected, analyzed, and interpreted biostratigraphic data and worked with asset stakeholders to ensure that research progress continued within the agreed upon scope of work. G. Gregory and J. Oefinger processed samples and collected geochemical data. G. Sharman prepared the manuscript with contributions from all co-authors.

**Competing interests**

The authors declare that they have no conflict of interest.

**Acknowledgments**

We thank the Chevron Gulf of Mexico Business Unit for access to core, scientific conversations, and the approval to publish these data. We thank the Chevron Technology Center for biostratigraphy and geochronology strategic research funding and

support of the University of Arkansas Detrital Geochronology Laboratory (DGL). Erin Meyers and Ellen Reat Wersan provided useful reviews of early manuscripts. Erik Pollock at the University of Arkansas Stable Isotope Laboratory (UASIL) assisted





with laboratory analyses. We thank Mac McGilvery for helpful discussions of Wilcox Group depositional facies. The views and conclusions expressed by the authors are solely their own and do not necessarily reflect the opinions, conclusions, or beliefs of Chevron Technology Center or its affiliates.

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
