# Peer review of "Carbon-isotope chemostratigraphy, geochemistry, and biostratigraphy of the Paleocene-Eocene Thermal Maximum, deepwater Wilcox Group, Gulf of Mexico (U.S.A.)"

_Climate of the Past, 2022_

## Author Comment (AC1)

Our Responses are shown in **bold, red text**

Reviewer #1

Sharman et al. present a new and very important PETM record within deep-water strata of the Gulf of Mexico within this manuscript. Data include a new bulk organic d13C record and associated TOC record, palynology, nannofossils/forams, and a detailed sedimentologic history. Data spanning the PETM from the Gulf of Mexico has been of long-standing interest to the broader Paleogene community and this is a welcome contribution. Broadly, they document a 2 per mille decrease in d13C that corresponds with the biostratigraphic Paleocene-Eocene boundary. This negative carbon isotope excursion coincides with a decrease in TOC, increase in terrestrial palynomorphs, a shift towards finer grained deposition, and reduced bioturbation. This pattern bears a resemblance to several other PETM sections globally and suggests greater fine-grained sediment flux from continents and a shift to less oxic conditions. The manuscript is well-written, clear, and concise. Figures are quite good. My main comments surround some of the interpretations of the data, though even these are not major. Below I have separated them by topic.

Line 23: Would the increase in CaCO3 post-dating the PETM be more consistent with increased limestone deposition as a carbon sequestration mechanism? While there is evidence for ocean acidification and shoaling of the CCD illustrated by the Zachos et al. 2005 Science paper (and others) this was occurring in deeper water overall it seems. From the description of the GOM data it sounds as though carbonate was not particularly abundant in the late Paleocene/early Eocene and it is really just a spike post/late-PETM. If Wilcox Group strata examined here are shallower than 2000 m, the CCD might not have shoaled that high? And in such a case perhaps the increase is actually the pulse of carbonate deposition seen in other sections.

**Our Response: This is a fair point. Given a lack of primary data on water acidity in our dataset, we have deleted the last sentence of the 1st paragraph in the abstract to avoid emphasis on the CCD position. We also now include a statement that allows for the possibility of enhanced carbonate deposition in Unit C reflecting coccolithophore blooms (citing Kelly et al., 2005: *Paleoceanography*), thus contributing to a global pattern of $CO_2$ drawdown via carbonate deposition. We have updated the Discussion section 5.1.1. to cite the interpretation for early dissolution within the lower part of the main CIE in the Logan-1 well (citing Vimpere et al., 2023: *Geology*), which is consistent with the lack of $CaCO_3$ in the early phases of the CIE in the Anchor 3 well. However, the Logan-1 well is some 150 km distant and more distal relative to our locality, thus presumably at a greater water depth.**

Line 212-224: Removal of hydrocarbon material via the solvent extraction method is beyond my area of expertise. To an outsider, this seems like an appropriate approach, but again I am not an expert. However, this is a key sample treatment technique that needs to be 100% certain since the geologic and climatic interpretations hang on an accurate d13C curve.

**Our Response: We now cite other studies that have used solvent extraction to remove petroleum contamination prior to d13C analysis. We also now include two additional**

**supplemental tables, one that illustrates the overall efficiency of oil contamination removal (Table S2 in the new submission) and one that provides comparisons of %C and d13C values for raw, solvent extracted, and solvent extracted + decarbonated samples as part of a pilot study conducted in the early phases of the research (Table S3 in the new submission). See also response to Reviewer #2.**

Table 1: Lf-1 should be LF-1 to maintain consistency with text.

**Our Response: We have fixed this.**

The authors invoke a sea-level rise as a needed contributor to sequestering coarse-grained material in proximal marginal marine environments, while export of fine-grained component and associated terrestrial palynomorph were able to deposit in deep-water. Is there sedimentologic evidence for a short-lived transgression within the GOM at the P-E boundary? It seems that if nonmarine basins are preferentially storing coarser sediment loads in North America that this phenomenon in a of itself might be sufficient to cause a shift towards finer grained deposition in deep-water from a mass balance perspective. This assumes a similar grain size distribution of sediment within the routing system before, during, and after the PETM. Do the authors have thoughts/opinions on this hypothesis?

**Our Response: We now cite Sluijs et al. (2008b) and Sluijs et al. (2014) in the second paragraph of Discussion section 5.4 as supporting our interpretation of sea-level rise. Sluijs et al., 2014 provides evidence for sea-level rise in the Gulf of Mexico specifically (Harrel core) and Sluijs et al. (2008b) reviews evidence for sea-level rise more globally.**

**It is an interesting question of whether coarser grained sediment was preferentially stored in onshore basins versus being exported to the coastline. Two additional PETM localities in the Gulf Coast (eastern Texas, Wilcox and Claiborne groups) have been proposed by Sharman et al. (http://dx.doi.org/10.2139/ssrn.4200185). These authors interpret the PETM to coincide with the basal, sand-rich Carrizo Formation, suggesting that coarser-grained sediment did reach the deltaic centers of eastern Texas (versus being sequestered inland).**

Several studies invoke a shift in oxygen availability during the PETM that caused a reduction in bioturbation. However, it seems to me this also could result if sedimentation rates increased and biologic disturbance remained constant. In the case of greater evidence for terrestrial fine-grained sediment export this seems like a possibility to evaluate in the manuscript's discussion.

**Our Response: This is a fair point. We now include a statement that allows for this possibility in section 5.3.**

I think the authors may have over-interpreted the CIA shifts observed, particularly given uncertainties around the CaO. I think the index is too insensitive to evaluate whether a pulse of fresh source material has been provided.

**Our Response: We have deleted the two paragraphs in Section 5.1.4. that interpreted the timing of the lag associated with changing CIA values (see also Reviewer #2 comment). We**

**have also deleted a sentence in the Conclusions pertaining to interpreting the CIA data as being in response to exhumation. We have also added text and references that state the influence of grain size, provenance, and carbonate/phosphate minerals on influenced CIA, in addition to silicate weathering (top of section 5.1.3).**

**Although we have deemphasized aspects of our interpretation of the CIA index, we do hold that the shift in CIA values from Paleocene to Eocene cannot be easily explained by increases in calcite or other carbonate minerals. This is because our XRD data do not show increased calcite content in Paleocene vs Eocene samples (excluding anomalously cemented or marly samples of Unit C), and biostratigraphic data do not indicate elevated counts of calcareous nannoplankton above the marly portion of the PETM. That said, we agree that additional research would be beneficial for exploring additional factors that may be influencing CIA values in early Eocene samples, outside of the effects of changes in silicate weathering.**

Final thought, I think the authors could be slightly more conservative with the sequence of events and the overall structure of the PETM release, body, recovery. My concern is that for some data (Fig 4) there are not that many data points within the CIE (3-5) which makes interpreting the finer structure of response difficult and uncertain. I think it is entirely reasonable to treat the PETM CIE zone as a whole and discuss average responses until such a point when more samples and analyses are performed. This is also dependent on the fidelity and precision of the d13Corg data, which are notoriously noisy. I am not insisting more analyses are performed. The work is certainly comprehensive, and I think a more conservative interpretation may increase the impact of the work.

**Our Response: We now add an additional statements in section 5.1.2 that highlight the uncertainty in the boundary between the main CIE and CIE recovery, given the relative noisiness of the d13C isotopic excursion. Vimpere et al. (2023) also noted uncertainty in the transition from main CIE to CIE recovery in the Logan-1 well.**

**Although it is true that the density of our biostratigraphic data, which were conducted for routine industry application, is limited over the relatively thin PETM interval, our lithologic, C-isotopic, and geochemical data were collected at much higher resolution (Figs. 4 and 7). For this reason, our interpretation of the biostratigraphic data is focused on the CIE zone as a whole, as suggested here (e.g., sections 4.4 and 4.5). However, we believe that a 4-fold division of the PETM interval is warranted given the density of lithologic and geochemical data (Fig. 7), particularly since a similar pattern of lithologic change was found by Vimpere et al. (2023) in the Logan-1 well. We now include additional statements in section 5.1.1 that compare the Anchor 3 core with the Logan-1 well as recently presented by Vimpere et al. (2023): *Geology*.**

---

## Author Comment (AC2)

Our responses are shown in **bold, red text**

Reviewer #2

**General Comments**

Sharman et al. present a wealth of high-quality datasets, including the important Corg CIE, Apectodinium, and Rhomboaster, which constrain the PETM in the deepwater Gulf of Mexico in the Anchor 3 well and allow details of the biotic and geochemical changes through the event to be evaluated. The deep-water Anchor 3 location is crucial for evaluating sediment supply characteristics and timing in large-scale routing systems, especially for rapidly deposited, mud-silt turbidites in distal marine PETM sections. Furthermore, their interpretations strengthen the hypothesis that the Gulf of Mexico (GoM) was open and hydraulically connected to the Atlantic before, during, and after the PETM despite potential blockage by collision of the Cuban and North American Plates. This manuscript is well written with datasets, assumptions, and interpretations clearly described and explained. Figures and captions are clear and well annotated.

**Specific Comments**

Line 205 – 235 - The overall biozonation and age model and co-occurrence of Apectodinium, Rhomboaster, and CIE provide convincing evidence for the PETM, main body and recovery intervals at the Anchor well. However, more scrutiny of the main d13Corg CIE is needed given the very organic-lean nature of the PETM interval dominated by coaly and inertinitic kerogen. This may indicate that it contains recycled organic material from older onshore sections. To what degree is this CIE being influenced by terrestrial or even recycled terrigenous kerogen and not recording the global exogenic carbon cycle (as in Sluijs and Dickens, 2012)?

**Our Response: This is a good consideration. We now include a statement in section 5.1.2 that makes a link between the degraded nature of the kerogen with the overall noisiness of the d13C isotopic excursion, citing Sluijs and Dickens (2012) and Aze et al. (2014). We have also added a statement in section 5.1.2 that notes the uncertainty in the boundary between our interpretations of the main CIE and CIE recovery.**

Also, what is the effectiveness of the extraction/clean-up procedure to remove invaded petroleum prior to d13Corg and TOC analysis? A further explanation of the amount of contamination noted and pre- and post-extraction TOC differences would help improve confidence in the results.

**We now provide an additional supplemental table (Table S3 in the revised submission) that includes data on how TOC values vary by treatment (raw, solvent extracted, and solvent extracted + decarbonated). Table S3 shows that TOC values do not substantially decrease in mudstone samples following solvent extraction, unlike in sandstones which are clearly oil stained. We also provide an appendix (Table S2) that describes the result of an experiment where the solvent extraction was shown to be ~99.998% effective at removing petroleum contamination (two-stroke motor oil) applied to standards.**

Line 503-507 - Sharman et al. propose a flood of TOC-poor terrigenous clay as an explanation for the remarkable TOC decrease at Anchor in Units B and C of the PETM. This organic lean clay is suggested to have been sourced from terrestrial environments undergoing enhanced oxidation of paleosols perhaps 100s of km from the well location given the ~31 ky time lag from PETM onset to lithologic response (Line 565). It seems important to also recognize the enlarging neritic mud apron associated with sea-level rise as the main or intermediate source regions for the low TOC clay. Drops in palynomorph and phytoclast abundance, oxidation of organics, and longer sediment residence times may have occurred there and further decreased TOC in the clay fraction. The time lag may reflect the reorganizing and mobilizing of muds in the submarine segment of the routing system.

**Our Response: We now include a statement that allows for the possible contributions of organic matter degradation within the shallow marine environment in section 5.1.2.**

Line 569-574 - The shale and sandstone CIA signals appear to become noisy during the CIE making the placement of clear offsets and estimated lags questionable. Also, CaO continues to increase through the Eocene after the PETM carbonate pulse suggesting carbonate minerals may be affecting the later shale CIA readings. It would be better to recognize the interpreted increasing influence of erosional denudation in the catchment through and following the PETM without being too explicit with the timing.

**Our Response: We have deleted the two paragraphs that interpreted the lag time associated with changes in CIA in section 5.1.4. We agree that the resolution on the lag-time is a bit noisy given variability observed around the PETM. We instead focus our interpretation on the general change in CIA values from late Paleocene to early Eocene time in section 5.1.3.**

Line 663-676 - The drop in TOC beginning in Unit A and mainly affecting Units B and C of the main CIE is incongruous with evidence for deoxygenation including lack of bioturbation and loss of benthic foraminifera. The ICPMS results in the supplement for Anchor do not show U and V enrichments and other redox-sensitive ratios in the CIE which along with low TOC suggest only weak benthic deoxygenation may have existed at Anchor. Given this, preservation of marine organics would have been diminished and high siliciclastic sedimentation rates (Line 508) should have not led to a dilution penalty on TOC in Units B and C. New data published in Vimpere et al. (2023) for the Logan well which is more distal than Anchor also show a drop in TOC over the clay-rich main CIE. However, the TOC decrease is not as severe as at Anchor despite a much higher sedimentation rate over the CIE. The differing decreases in TOC through the main CIE at Anchor and Logan suggest that although the influx of organic-lean clay was broadly distributed, depositional and organic preservational conditions varied over the GOM fan system.

**Our Response: We now include discussion of U, V, and Mo concentrations from ICP-MS and provide a supplementary table with the aluminum-normalized values and the enrichment factors for Mo and U. The concentrations of redox sensitive trace elements such U, V, and Mo are either low or below detection limit, which may be consistent with suboxic conditions, but do not provide evidence of significant anoxia.**

**We also now include a comparison with organic carbon values in the Logan well (Vimpere et al. (2023): *Geology*) in our discussion section 5.1.2. As noted by Reviewer #2, Vimpere et al. (2023) observed a decrease of ~0.6% from late Paleocene to the PETM CIE, which is less pronounced than we found in our core samples. However, we are uncertain to what extent these differences may reflect differences in sample type (ditch-cuttings vs core plugs), location, or other factors.**

Section 5.4 - This important section is too heavy on literature discussion and needs more focus on the Anchor routing system, specifically the transition from terrestrial to marine segments of the routing system. The PETM seal-level rise presumably led to the growth of a mud-rich apron in the transition region. Did clays and organics from terrestrial environments continue to follow the long-lived routing system through this region or did a modified marine routing system develop?

**Our Response: We now include a reference to a study that is in revision to Palaeo3 (Sharman et al., http://dx.doi.org/10.2139/ssrn.4200185) which is relevant to the terrestrial-to-marine segment of the Wilcox sediment routing system. However, given the unpublished nature of this work, we prefer to put our results in context with other better-documented PETM localities that span onshore to offshore setting (e.g., Fig. 10). We agree with the general point made here that it would be helpful to identify additional Paleocene-Eocene boundary sections within the greater Wilcox system to better understand sediment transport. However, it is worth noting that much of the Wilcox between the relatively shallow, onshore (fluvio-deltaic) and distal, deep-water sections is buried very deeply beneath younger Cenozoic fill of the northern Gulf of Mexico.**

Finally, I think the length and level of detail in this manuscript are warranted given the size and scope of the datasets. However, the impact of this manuscript will be increased if the Logan well results are integrated into the discussion. This may expand the understanding of organic-lean clay injection and spatial variations in deoxygenation and marine versus terrestrial organic matter supply over the PETM in the deep-water GoM.

**Our Response: We now include references to Vimpere et al. (2023) throughout the revised manuscript, as suggested.**

Vimpere, L., et al., 2023, Carbon isotope and biostratigraphic evidence for an expanded Paleocene–Eocene Thermal Maximum sedimentary record in the deep Gulf of Mexico: Geology, v. XX, https://doi.org/10.1130/G50641.1

**Text and Figure Corrections**

**Our Response: We have made all suggested changes to text and figures below.**

Figure 2 – change Suwannee Straight to Strait

Table 1 – LF-1 vs Lf-1

Line 141 – Add core depths in meters.  Given the use of feet and meters in the text, the well data profiles would benefit from having the depth scale provided in meters as well as feet instead of having just a dual scale bar in Figures 4-7.

Line 516 -  …to record a transition from low TOC…

Line 533 – hydrologic vs hydraulic cycle?

Line 552 – finger-grained to finer-grained

Line 553 – coaster to coarser

Line 773 – coaster to coarser